# Improving the Unsupervised Disentangled Representation Learning with VAE Ensemble

## Abstract

Variational Autoencoder (VAE) based frameworks have achieved the state-of-the-art performance on the unsupervised disentangled representation learning. A recent theoretical analysis shows that such success is mainly due to the VAE implementation choices that encourage a PCA-like behavior locally on data samples. Despite this implied model identifiability, the VAE based disentanglement frameworks still face the trade-off between the local orthogonality and data reconstruction. As a result, models with the same architecture and hyperparameter setting can sometimes learn entangled representations. To address this challenge, we propose a simple yet effective VAE ensemble framework consisting of multiple VAEs. It is based on the assumption that entangled representations are unique in their own ways, and the disentangled representations are "alike" (similar up to a signed permutation transformation). In the proposed VAE ensemble, each model not only maintains its original objective, but also encodes to and decodes from other models through pair-wise linear transformations between the latent representations. We show both theoretically and experimentally, the VAE ensemble objective encourages the linear transformations connecting the VAEs to be trivial transformations, aligning the latent representations of different models to be "alike". We compare our approach with the state-of-the-art unsupervised disentangled representation learning approaches and show the improved performance.

## 1 Introduction

Disentangled representation learning aims to capture the semantically meaningful compositional representation of data (Higgins et al., 2018; Mathieu et al., 2018), and is shown to improve the efficiency and generalization of supervised learning (Locatello et al., 2019), reinforcement learning (Watters et al., 2019), and reasoning tasks (van Steenkiste et al., 2019). The current state-of-the-art unsupervised disentangled representation learning deploy the Variational Autoencoder (VAE) (Kingma & Welling, 2013; Rezende et al., 2014). The main challenge is to reduce the trade-off between learning a disentangled representation and reconstructing input data. Most of the recent works extend the original VAE objective with carefully designed augmented objective to address this trade-off (Higgins et al., 2017; Burgess et al., 2017; Kim & Mnih, 2018; Chen et al., 2018; Kumar et al., 2017). A recent study in (Locatello et al., 2018) compared these methods and showed that their performance is sensitive to initialization and hyperparameter setting of the augmented objective function.

Recently, Duan *et al.* (Duan et al., 2019) developed an unsupervised model selection method named Unsupervised Disentanglement Ranking (UDR) to address the challenge of hyperparameter search and model selection. UDR leverages the finding in (Rolinek et al., 2019) that the implementation choices of VAE encourage a local PCA-like behavior locally on data samples. As a result, disentangled representations by VAEs are "alike" as they are similar up to signed permutation transformations. On the contrary, the entangled representations by VAEs are "unique" as they are similar at least up to non-degenerate rotation matrices. UDR uses multiple models trained with different initializations and hyperparameter settings, and builds a similarity matrix measuring the pair-wise similarity between the latent variables from different models. A higher score is given to the model that can match its representations to many others models. The results show close match between UDR and commonly used supervised metrics, as well as the performance of downstream tasks using the latent representations.

Inspired by the findings from these studies, we propose a simple yet effective VAE ensemble framework to improve the disentangled representation by VAE. The proposed VAE ensemble consists of multiple VAEs. The latent variables in every pair of these models are connected through linear layers to force the latent representations in the ensemble to be similar up to a linear transformation. We show that the VAE ensemble objective encourages these pair-wise linear transformations to converge to trivial transformations, making latent representations of different VAEs in the ensemble to be "alike", thus disentangled. In this paper, we make the following contributions: (1) We introduce a simple yet effective VAE ensemble framework to improve the disentangled representation learning using the original VAE. (2) We show in theoretical analysis that the linear transformations connecting the latent representations of the individual models in the ensemble tend to converge to trivial transformations thus encourage disentangled representation, and verify this result with experiments. (3) We evaluate our approach using the original VAE model, and show the improved state-of-the-art performance across different datasets.

## 2 RELATED WORK

**Variatioanl Autoencoder** is a deep directed probabilistic graphical model consisting of an encoder and a decoder (Kingma & Welling, 2013; Rezende et al., 2014). The encoder $q_\phi(z|x)$ maps the input data $x \in \mathbb{R}^n$ to a probabilistic distribution as the latent representation $z \in \mathbb{R}^d$, and the decoder $q_\theta(x|z)$ maps the latent representation to the data space noted as $q_\theta(x|z)$, where $\phi$ and $\theta$ represent model parameters. The VAE objective is to maximize the marginalized log-likelihood of data. Direct optimization of this objective is not tractable and it is approximated by the evidence lower bound (ELBO) as:

$$\mathcal{L}_{VAE} = \mathbb{E}_{q_\phi(z|x)}[\log q_\theta(x|z)] - \text{KL}(q_\phi(z|x) \parallel p(z)), \tag{1}$$

In practice, the first term is estimated by reconstruction error. The second term is the Kullback-Leibler divergence between the posterior $q_\phi(z|x)$ and the prior $p(z)$ commonly chosen as an isotropic unit Gaussian $p(z) \sim \mathcal{N}(0, \mathbf{I})$.

**Disentangled representation by VAE** has achieved the state-of-the-art performance (Higgins et al., 2017; Burgess et al., 2017; Kim & Mnih, 2018; Chen et al., 2018; Kumar et al., 2017), despite the fact that the VAE objective only models the marginal distribution of the data instead of the desired joint distribution over data and latent variables. The reason for this success is the implementation choices of the VAE framework (Rolinek et al., 2019). In practice, the latent variables in VAE often work in "polarized" modes. The "passive" mode is defined by $\mu_j^2(x) \ll 1$ and $\sigma_j^2(x) \approx 1$, while the "active" mode is defined by $\sigma_j^2(x) \ll 1$. The "passive" latent variables closely approximate the prior and have little effect on the decoder. The "active" latent variables, on the other hand, are closely related to both the per sample KL loss and the decoder output. The "polarized regime" enables a reformulated VAE objective showing that VAEs optimize a trade-off between data reconstruction and orthogonality of the linear approximation of decoder Jacobian locally around a data sample. This PCA-like behavior near data points encourages an identifiable disentangled latent space by VAE. Furthermore, it was suggested that finding an appropriate "polarized regime" is dependent on the initialization and the hyperparameter tuning of the state-of-the-art approaches. In this study, we show that the proposed VAE ensemble aligns the "polarized regime" of individual VAE models towards the disentangled representation.

**Model selection** In practice, we often observe neural networks achieve similar performance with different internal representations when trained with the same hyperparameters (Raghu et al., 2017; Wang et al., 2018; Morcos et al., 2018). For the unsupervised disentanlged representation, as discussed in (Locatello et al., 2018; Duan et al., 2019), we often observe high variance in the performance from the model trained with the same architecture and hyperparameter setting. This poses a challenge for choosing the model in practice. Duan et al. (2019) proposed Unsupervised Disentanglement Ranking (UDR) to address this challenge. The extensive empirical evaluations on UDR using both the supervised metric measurement and the performance of downstream tasks validates its effectiveness. They also confirm that disentangled representations are "alike" and entangled representations are unique in their own ways. The proposed VAE ensemble leverages this finding.

**Identifiable VAE** Built on the recent breakthroughs in nonlinear Independent Component Analysis (ICA) literature (Hyvarinen & Morioka, 2016; 2017; Hyvarinen et al., 2019), Khemakhem *et al.* show that the identification of the true joint distribution over observed and latent variables is

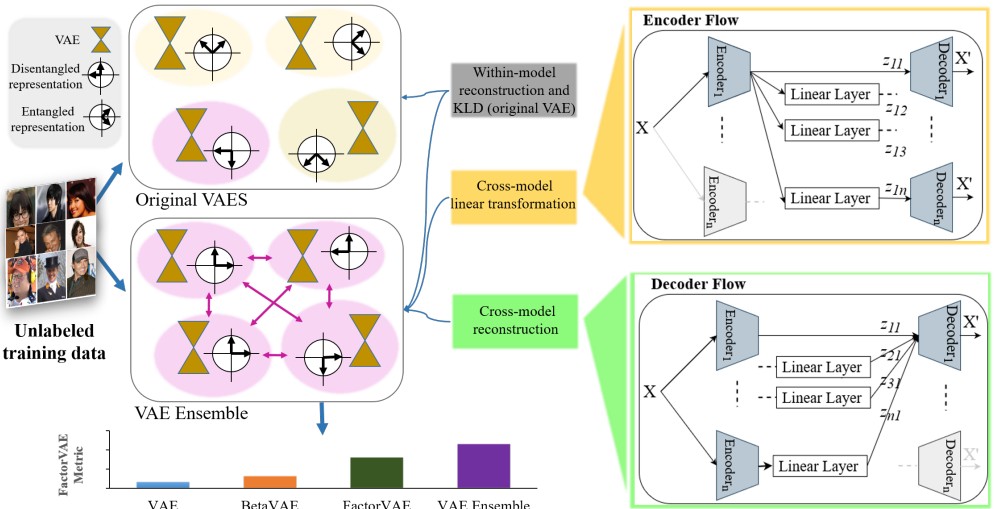

Figure 1: The proposed VAE ensemble consists of multiple original VAE models. The encoders of the VAEs in the ensemble generate input encoding that can be linearly transformed among each other. The decoders of the VAEs in the ensemble reconstruct the input data from both their corresponding encoder and the linearly transformed encodings from other encoders. The $x$ and $y$ axis of the circles on the left hand side of the plot represent two generative factors as an example. The aligned arrows with $x$ and $y$ axis show a model with disentangled representation and unaligned ones show a model with entangled representation.

possible up to very simple transformations (Khemakhem et al., 2019). They proposed identifiable VAE (iVAE) that requires a factorized prior distribution over the latent variables conditioned on an additionally observed variable, such as a class label or almost any other observation. We believe the proposed VAE ensemble is related to such framework where the latent representation from one VAE model can be regarded as the auxiliary observations for another.

**Ensemble learning** The idea of ensemble learning is to combine multiple learning models (potentially weak learners) to improve the task performance or robustness over a single model (Schapire, 1990). It achieves the improved performance by averaging the bias, reducing the variance thus preventing the over-fitting (Drucker et al., 1994; Breiman, 1996). Early works in neural networks have used the ensemble learning to achieve top performance in the related competition (Krizhevsky et al., 2012; Simonyan & Zisserman, 2014). In this work, we apply the ensemble learning to enforce the alignment among the latent representations of different models. This results in latent representations that are similar among each other in the ensemble up to a trivial transformation.

## 3 THE VAE ENSEMBLE FRAMEWORK

As illustrated in Figure 1, the proposed VAE ensemble consists of $n$ original VAE models with the same architecture but different initializations. It also consists of $n \times (n-1)$ linear layers connecting the latent representations of every two VAE models. Each model in the ensemble maintains its original VAE objective as Eq. 1. In addition, $l_2$ loss is used to force mapping between latent representations via pair-wise linear layers (cross-model linear transformation). The decoder of each VAE model generates the input reconstruction from not only their corresponding encoder (within-model reconstruction), but also the linearly transformed encodings from other encoders (cross-model reconstruction). Overall, the VAE ensemble is trained with the following objective:

$$\mathcal{L}(\boldsymbol{\theta}, \boldsymbol{\phi}) = \sum_{i=1}^{n} \sum_{j=1}^{n} \mathbb{E}_{q_{\phi ij}(z_{ij}|x)}[\log q_{\theta j}(x|z_{ij})] - \sum_{i=1}^{n} \mathrm{KL}(q_{\phi ii}(z_{ii}|x) \parallel p(z_{ii}))$$
$$- \gamma \sum_{i=1}^{n} \sum_{j=1}^{n} \mathbb{E}_{q_{\phi ij}(z_{ij}|x)} \|z_{jj} - z_{ij}\|^2, \tag{2}$$

where $n$ is the number of models in the ensemble; $\phi := (\phi_{ij})$ is the encoders parameters where $\phi_{ij}$ represents the encoder of $VAE_i$ and its associated linear layer mapping the latent representation from $VAE_i$ to $VAE_j$ (notice that $\phi_{ii}$ represents the encoder parameters of $VAE_i$ only and its associated linear transformation can be be regarded as an identity transformation); $\theta := (\theta_i)$ represents the decoder parameters of $VAE_i$; and $z_{jj}$ represents the latent representation of $VAE_j$ while $z_{ij}$ represents the linearly transformed latent representation from $VAE_i$ to $VAE_j$. $\gamma$ is a hyperparameter to balance the effect of the estimation error between the latent representaitons. $p(z_{ii}) \sim \mathcal{N}(0, I)$ is assume to be the prior as defined in the original VAE objective.

Comparing to the original VAE objective in Eq. 1, the objective of each individual VAE model in the ensemble, $\mathcal{L}'_{VAE}$, can be written as:

$$\mathcal{L}'_{VAE}(\theta, \phi) = \mathcal{L}_{VAE}(\theta, \phi) + \underbrace{\sum_{j=1}^{n-1} \left\{ \mathbb{E}_{q_{\phi_j}(z_j|x)}[\log q_\theta(x|z_j)] - \gamma \mathbb{E}_{q_{\phi_j}(z_j|x)} \|z_j - z\|^2 \right\}}_{Ensemble\ Regularization}, \quad (3)$$

where $n$ is the number of models in the ensemble, $\phi_j$ stands for the parameters of the encoder and its linear transformation layers from other VAEs, and $z_j$ represents the linear transformed latent representation of the encoding from other encoders.

In this form, the VAE ensemble regularizes each VAE model with additional terms on the encoder as $\gamma \mathbb{E}_{q_{\phi_j}(z_j|x)} \|z_j - z\|^2$, and on the decoder as $\sum_{j=1}^n \mathbb{E}_{q_{\phi_j}(z_j|x)}[\log q_\theta(x|z_j)]$. These regularizations directly constrain the latent representations among different VAE models in the ensemble to be similar. In particular, for a given input data, $\|z_j - z\|^2$ encourages the encoders to generate similar encodings up to the linear transformations; and $\sum_{j=1}^n \mathbb{E}_{q_{\phi_j}(z_j|x)}[\log q_\theta(x|z_j)]$ emphasizes the similar effect on the data reconstruction from the latent variables such that the decoders can reconstruct the input data with both the original encoding $z$ and the linearly transformed encoding $z_j$. As we shall discuss in the next section, together these regularizations encourage similar latent representation up a trivial transformation by different models in the ensemble.

We also introduce the hyperparameter $\gamma$ to balance the trade-off between these two regularizations: higher value forces closer mapping between the encoders and reduce the cross-model reconstruction error of the decoders; lower value relaxes the mapping between the encoders and increases the cross-model reconstruction error of the decoders. As we show in Section 5, both components are important and balancing the trade-off between them is important as the ensemble size increases.

**Computational complexity** It is a common practice to train a number of seeds per hyperparameter setting for the current state-of-the-art unsupervised disentanglement VAE models (Locatello et al., 2018; Duan et al., 2019). Comparing to training $n$ original VAEs, the proposed VAE ensemble requires additional $n \times (n-1)$ linear layers. While this addition does not increase the size of the model much, the estimation of the linear transformations loss and the cross-model reconstruction losses grow with $n \times (n-1)$, which may be computationally expensive especially when $n$ is large. That being said, the results in Section 5 show that the VAE ensemble achieves more stable results comparing to the current state-of-the-art models. Also, its computation is highly parallelisable.

## 4 THEORETICAL JUSTIFICATION

In this section, we present the theoretical analysis on why the proposed VAE ensemble can improve the disentangled representation. We start with analysing the $l_2$ objective in Eq. 2 of the pair-wise liner transformations in the VAE ensemble, and show that: (1) the pair-wise linear transformations encourage similar "polarized" regime (see Sec. 2) among the VAEs in the ensemble; (2) the linear transformations are close to the orthogonal transformations. Based on these two properties, we then discuss how the cross-model reconstructions by the VAE ensemble encourage learning a disentangled representation over its entangled counterpart.

### 4.1 THE EFFECT OF LINEAR TRANSFORMATION BETWEEN LATENT REPRESENTATIONS

Let $VAE_i$ and $VAE_j$ be two different VAE models in the ensemble, and $M_{ji}$ be the linear transformation that maps the latent representation of a given input $x$ by $VAE_j$ to the one by $VAE_i$, as

$\mathbf{z_j}(x) \sim \mathcal{N}(\boldsymbol{\mu}_j(x), diag(\boldsymbol{\sigma}_j(x)^2))$ to $\mathbf{z_i}(x) \sim \mathcal{N}(\boldsymbol{\mu}_i(x), diag(\boldsymbol{\sigma}_i(x)^2))$. In the following we remove the input notation from the VAE latent representations for simplicity (i.e. $\mathbf{z_j}(x)$ is simplified as $\mathbf{z_j}$), while keeping in mind that the analysis is based on the local latent representation of a given input $x$.

For $\text{VAE}_j$, the $l_2$ term of the VAE ensemble loss in Eq. 2 aims to find $M_{ji}$ and $\mathbf{z_j}$ that minimize $\mathbb{E} \|\mathbf{z_i} - M_{ji}\mathbf{z_j}\|^2$, where the expectation is over the stochasticity of $\text{VAE}_j$. We can write $\mathbf{z_i}$ and $\mathbf{z_j}$ as $\mathbf{z_i} = \boldsymbol{\mu}_i + \boldsymbol{\epsilon}_i$ and $\mathbf{z_j} = \boldsymbol{\mu}_j + \boldsymbol{\epsilon}_j$, where $\boldsymbol{\epsilon}_i \sim \mathcal{N}(0, diag(\boldsymbol{\sigma}_i^2))$ and $\boldsymbol{\epsilon}_j \sim \mathcal{N}(0, diag(\boldsymbol{\sigma}_j^2))$. Hence using bias-variance decomposition, the $l_2$ term can be written as:

$$
\begin{aligned}
&\min_{M_{ji}, \mathbf{z_j} \sim \mathcal{N}(\boldsymbol{\mu}_j, diag(\boldsymbol{\sigma}_j^2))} \mathbb{E} \|\mathbf{z_i} - M_{ji}\mathbf{z_j}\|^2 \\
&= \min_{M_{ji}, \mathbf{z_j} \sim \mathcal{N}(\boldsymbol{\mu}_j, diag(\boldsymbol{\sigma}_j^2))} \left\{ \|\boldsymbol{\mu}_i - M_{ji}\boldsymbol{\mu}_j\|^2 + \mathbb{E}_{\mathbf{z_j}} \|M_{ji}\boldsymbol{\mu}_j - M_{ji}\mathbf{z_j}\|^2 + \mathbb{E}_{\mathbf{z_i}} \|\boldsymbol{\mu}_i - \mathbf{z_i}\|^2 \right\} \\
&= \min_{M_{ji}, \mathbf{z_j} \sim \mathcal{N}(\boldsymbol{\mu}_j, diag(\boldsymbol{\sigma}_j^2))} \left\{ \|\boldsymbol{\mu}_i - M_{ji}\boldsymbol{\mu}_j\|^2 + \mathbb{E}_{\mathbf{z_j}} \|M_{ji}\boldsymbol{\mu}_j - M_{ji}\mathbf{z_j}\|^2 \right\} + C_1 \\
&= \min_{M_{ji}, \mathbf{z_j} \sim \mathcal{N}(\boldsymbol{\mu}_j, diag(\boldsymbol{\sigma}_j^2))} \left\{ \|\boldsymbol{\mu}_i - M_{ji}\boldsymbol{\mu}_j\|^2 + \mathbb{E}_{\boldsymbol{\epsilon}_j} \|M_{ji}\boldsymbol{\epsilon}_j\|^2 \right\} + C_1
\end{aligned}
\tag{4}
$$

where the constant $C_1$ arises from the fact $\mathbb{E}_{\mathbf{z_i}} \|\boldsymbol{\mu}_i - \mathbf{z_i}\|^2$ does not depend on $M_{ji}$ and $\mathbf{z_j}$. Eq. 4 consists of a deterministic component of $\|\boldsymbol{\mu}_i - M_{ji}\boldsymbol{\mu}_j\|^2$ and a stochastic component of $\mathbb{E}_{\boldsymbol{\epsilon}_j} \|M_{ji}\boldsymbol{\epsilon}_j\|^2$.

The deterministic component can be minimized by adjusting the parameters in $\text{VAE}_j$ such that its mean encoding $\boldsymbol{\mu}_j$ is optimized for any given $M_{ji}$. This simplifies our analysis to focus on the stochastic component. Between $M_{ji}$ and $\boldsymbol{\epsilon}_j$ in this stochastic component, we separately optimize one while having the other fixed.

We start with fixed $M_{ji}$ and optimizing for $\boldsymbol{\epsilon}_j \sim \mathcal{N}(0, diag(\boldsymbol{\sigma}_j^2))$. Notice that $\boldsymbol{\sigma}_j^2$ is associated with $\text{VAE}_j$ objective. In (Rolinek et al., 2019), the VAE objective is reformulated into the deterministic reconstruction, the stochastic reconstruction and the KL loss. The last two components define the stochastic loss of VAE. It is formulated as:

$$
\min_{V, \boldsymbol{\sigma}_j^2} \sum_X \mathbb{E}_{\boldsymbol{\epsilon}_j \sim \mathcal{N}(0, diag(\boldsymbol{\sigma}_j^2))} \|D\boldsymbol{\epsilon}_j\|^2 \quad s.t. \quad \sum_X L_{KL} = C_1,
\tag{5}
$$

where $X$ represents the dataset, $D$ represents the local linear approximation of the Jacobian of the decoder with singular value decomposition as $D = U\Sigma V^T$. Furthermore, the KL loss $L_{KL} = \frac{1}{2} \sum_{k=1}^d (\mu_{jk}^2 + \sigma_{jk}^2 - \log \sigma_{jk}^2 - 1)$ can be simplified as $L_{\approx KL} = \frac{1}{2} \sum_{k \in \text{``active''}} (\mu_{jk}^2 - \log \sigma_{jk}^2 - 1)$ based on the "polarized" regime of VAE. (Rolinek et al., 2019) shows that $\boldsymbol{\sigma}_j^2$ act as the precision control allowed for each latent variable where more influential ones receive more precision. Combining the stochastic loss of the linear transformation in Eq. 4 and the stochastic loss of the original VAE in Eq. 5, the overall stochastic loss on $\boldsymbol{\sigma}_j^2$ can be formulated as:

$$
\min_{\boldsymbol{\sigma}_j^2} \mathbb{E}_{\boldsymbol{\epsilon}_j \sim \mathcal{N}(0, diag(\boldsymbol{\sigma}_j^2))} [\|M_{ji}\boldsymbol{\epsilon}_j\|^2 + \|D\boldsymbol{\epsilon}_j\|^2] \quad s.t. \quad \sum_k -\log \sigma_{jk}^2 = C_2,
\tag{6}
$$

where $\sigma_{jk}^2$ is the $k$th element of $\boldsymbol{\sigma}_j^2$. Here we further simplify $L_{KL}$ with the $L_{\approx KL}$ up to additive constants $C_2$ when $\boldsymbol{\mu}_j$ is fixed. In addition to the precision control of $\boldsymbol{\sigma}_j^2$ on $\text{VAE}_j$, this objective also aims to find an optimal distribution of $\boldsymbol{\sigma}_j^2$ that aligns the "polarized regime" among different VAEs. To see why, let $c_k$ be the $k$th column of $M_{ji}$, we then have $\mathbb{E} \|M_{ji}\boldsymbol{\epsilon}_j\|^2 = \sum_k \|c_k\|^2 \sigma_{jk}^2$. The Arithmetic-Mean–Geometric-Mean (AM/GM) inequality suggests that $\sum_k \|c_k\|^2 \sigma_{jk}^2 \geq n \left( \prod_k \|c_k\|^2 \sigma_{jk}^2 \right)^{1/n} = n \left( \prod_k \|c_k\|^2 \right)^{1/n} \exp(-C)$, where the equality is achieved when $\|c_m\|^2 \sigma_{jm}^2 = \|c_n\|^2 \sigma_{jn}^2$ for any $m \neq n$. This suggests that latent variables with high $\|c_k\|^2$ mapping from $\mathbf{z_j}$ to $\mathbf{z_i}$ will have smaller variance. Hence, these latent variables in $\mathbf{z_j}$ are encouraged to stay in the "active" mode. On contrary, the latent variables that do not share similar generative factors between $\mathbf{z_j}$ and $\mathbf{z_i}$ will be assigned larger variance, and being pushed towards the "passive" mode.

202 Now we fix the optimal distribution of $\boldsymbol{\sigma}_j^2$, and optimize for $M_{ji}$. Since $\boldsymbol{\epsilon}_j \sim \mathcal{N}(0, diag(\boldsymbol{\sigma}_j^2))$, this
203 objective can be understood as optimally rotating the latent space of $\text{VAE}_j$ such that the stochastic
204 component in Eq. 4 is minimized. Specifically, we have the following objective:

$$\min_{M_{ji}} \left\| M_{ji}\boldsymbol{\epsilon}_j \right\|^2 = \min_{R} \left\| M_{ji}R^T\boldsymbol{\epsilon}_j \right\|^2 \tag{7}$$

205 where $R$ is an orthogonal transformation. Let $c_k'$ be the $k$th column of $M_{ji}R$. Similar as before, the
206 AM/GM inequality suggests $\left\| M_{ji}R^T\boldsymbol{\epsilon}_j \right\|^2 = \sum_k \left\| c_k' \right\|^2 \sigma_{jk}^2 \geq \prod_k \left\| c_k' \right\|^2 \exp(-C_3)$. Hadamard's
207 inequality suggests that $\prod_k \left\| c_k' \right\|^2 \geq |\det(M_{ji}R)|$, and the equality is satisfied when $c_k'$s are pair-
208 wise orthogonal. This can be understood from the geometric perspective where $\prod_k \left\| c_k' \right\|^2$ gives an
209 upper bound on $\text{Volume}(\left\{ M_{ji}R^T x : x \in [0,1]^d \right\})$. As a result, the optimization of $M_{ji}$ will lead to
210 an orthogonal transformation. Together the optimization of Eq. 6 and Eq. 7 encourages the align-
211 ment of the "polarized regime" among different models under orthogonal linear transformations.
212 They force different models in the ensemble to capture the same mixture of the generative factors.
213 In the next section, we discuss the effect of the cross-model reconstruction in the VAE ensemble that
214 encourages the disentangled representation over the entangled ones.

## 4.2 THE EFFECT OF CROSS-MODEL RECONSTRUCTION

216 In an entangled representation, each latent variable captures a mixture of generative factors in its
217 unique way. Since different generative factors typically have different effects on data variations
218 (Duan et al., 2019), the orthogonal transformation from one entangled representation $\mathbf{z_j}$ to another
219 one $\mathbf{z_i}$ introduces different encoding variance. Some of the transformed latent variables in $M_{ji}\mathbf{z_j}$
220 carry larger variance comparing to the corresponding ones in $\mathbf{z_i}$. This discrepancy leads to larger
221 cross-model reconstruction of $\text{VAE}_i$ than the within model reconstruction. This error forces both
222 $\text{VAE}_i$ and $\text{VAE}_j$ to adjust their representations until the effect on the data reconstruction by indi-
223 vidual latent variables matches between $M_{ji}\mathbf{z_j}$ and $\mathbf{z_i}$. The process applies to all models in the
224 ensemble and eventually leads to a one-to-one mapping of latent variables between different mod-
225 els, where $M_{ji}$ becomes a trivial transformation (signed permutation matrix). In particular, if one of
226 the models in the VAE ensemble learns a disentangled representation, other models in the ensemble
227 will converge to it. This is because the orthogonal transformation from an entangled representation
228 to a disentangled representation introduces larger cross-model encoding variance due to the mixture
229 of different generative factors in the former, thus a larger cross-model reconstruction by the disen-
230 tangled model. On contrary, the orthogonal transformation from a disentangled representation to an
231 entangled representation would not introduce larger cross-model encoding variance than the within
232 model encoding, thus similar cross-model reconstruction as within model reconstruction by the en-
233 tangled model. Such a gap encourages the entangled representations to align with the disentangled
234 representation. We illustrate the geometric interpretation of such a case in Appendix C.

235 From these discussions, we conclude that the VAE ensemble encourages different individual mod-
236 els to capture similar generative factors, thus learn representations that are "alike" up to a trivial
237 transformation. In the next section, we verify these analytic results with experiments.

## 5 EXPERIMENTS

239 Our experiments are designed to confirm the discussions in the previous sections. Particularly we
240 ask the following questions: **Q1:** Do the linear transformations in the ensemble converge to trivial
241 transformation? **Q2:** Do the VAEs in the ensemble work in similar "polarized" regime? **Q3:** Does
242 VAE ensemble improve the unsupervised disentangled representation learning, and what is the ef-
243 fect of ensemble size? **Q4:** What are the effects of the cross-model reconstruction loss, the linear
244 transformation loss and the hyperparameter $\gamma$ in the VAE ensemble objective?

245 We analyze the inner working of the proposed VAE ensemble using the benchmark *dSprite* dataset
246 (Matthey et al., 2017) with fully known generative processes, and the real-world *CelebA* dataset (Liu
247 et al., 2015) with unknown generative process. Furthermore, for *dSprites* dataset, we compare our
248 results with the original VAE model and the state-of-the-art disentanglement VAE models including
249 $\beta$-VAE (Higgins et al., 2017), FactorVAE (Kim & Mnih, 2018), TC-VAE (Chen et al., 2018) and
250 DIP-VAE (Kumar et al., 2017). We use two widely used supervised metrics including FactorVAE
251 metric (Kim & Mnih, 2018) and DCI Disentanglement scores (Eastwood & Williams, 2018) as the

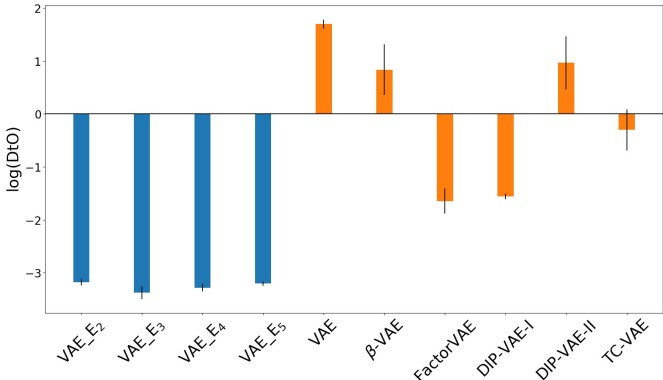

Figure 2: Comparing the DtO of linear transformations in the VAE ensemble ($\gamma$=10) with the one between well-trained individual VAEs, as well as the well-trained individual state-of-the-art VAE models. The latent dimension for all models is set to 10 and evaluated on the *dSprite* dataset.

quantitative measurements. They are shown to correlate with other common supervised metrics (Locatello et al., 2018). For example, FactorVAE metric and $\beta$-VAE metric (Higgins et al., 2017) capture similar notions, while DCI Disentanglement and Mutual Information Gap (MIG) (Chen et al., 2018) capture similar notions. In addition, DCI Disentanglement is closely related to the unsupervised model selection method UDR (Duan et al., 2019). For *CelebA* dataset, we show the latent traversal visulization as a qualitative measurement in Appendix E. We provide the details of the experiments in Appendix D.

**Q1:** We use the *Distance to Orthogonality* (DtO) (Rolinek et al., 2019) to check if the linear transformations in the ensemble converge to a signed permutation matrix during training. DtO is the Frobenius norm of the difference between a matrix $M$ and its closest signed permutation matrix $P(M)$. It is solved with mixed-integer linear programming (MILP) formulation. The details on DtO can be found in Appendix B. In Figure 2, we show the DtO estimation of the linear transformations in the VAE ensemble of different ensemble size for the *dSprite* dataset. We show the mean and standard deviation of DtO across all linear transformations over 10 different runs. Furthermore, we compare these results with a baseline measurement where DtO is estimated for the linear transformations between the mean latent representations of well-trained individual models. Specifically, we use ten well-trained individual models and report the mean and standard deviation of the DtO estimations. As seen in the figure, the VAE ensemble models with different ensemble size all approach to trivial transformations between the individual models, while other VAE models do not have such property. In Fig. 6, we show that during training, the VAE ensemble remains maintains low DtO while the original VAEs do not have such property. A similar result for models trained on the *CelebA* dataset with different latent dimensions is shown in Fig. 7. Further discussion on these results are provided in Appendix E.

**Q2:** To check if the models in the VAE ensemble work in similar "polarized" regime, we estimate the relative error between $L_{KL}$ and $L_{\approx KL}$ as $\Delta = \frac{L_{KL} - L_{\approx KL}}{L_{KL}}$ for each latent variable. Smaller $\Delta$ indicates closer matching between $L_{KL}$ and $L_{\approx KL}$ of a latent variable, thus more "active". Figure 3(a) and 3(b) show $\log(\Delta)$ of the 10 latent variables of individual models in VAE_$E_2$ and VAE_$E_3$ with different $\gamma$ settings trained on the *dSprite* dataset. The results show that individual models in the ensemble do work in similar "polarized regime'. In Figure 3(a), we also compare the VAE ensemble with the $\beta$-VAE where $\beta = 4$. This setting was found previously to be the optimal setting for the *dSprite* data for $\beta$-VAE (Higgins et al., 2017). We see that the VAE ensemble encourages more "active" latent variables than $\beta$-VAE. When we compare Fig. 3(a) and 3(b), we see that as the ensemble size increases, individual models are forced to have more "active" latent variables by decomposing the generative factors. This can be observed in the latent traversals shown in Appendix E. The *dSprites* dataset contains five ground truth generative factors. The VAE_$E_2$ models can have up to eight "active" latent variables depending on input, and these representations capture a decomposition of the ground truth generative factors.

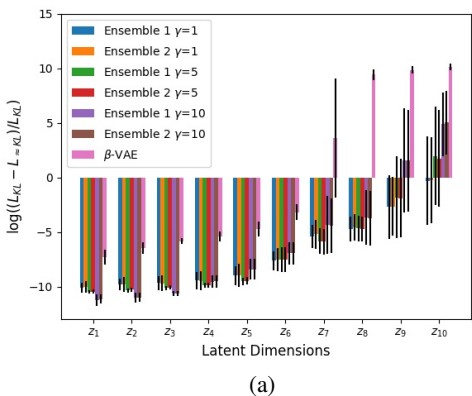 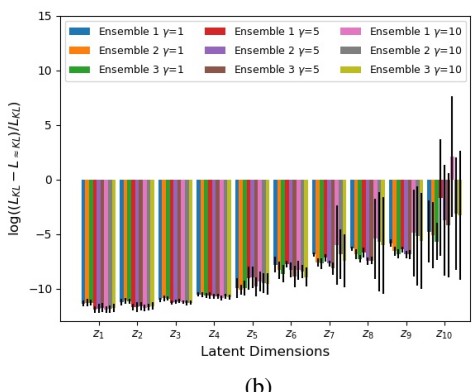

|     (a)     |     (b)     |

Figure 3: The "polarized regime" comparison between models in the VAE ensemble. The latent dimension is set to 10 and the results are over 10 runs of training on the *dSprite* dataset. (a) The "polarized regime" comparison by $\log(\frac{L_{KL}-L_{\approx KL}}{L_{KL}})$ of each latent variable of the two models in VAE_$E_2$ as well as $\beta$-VAE model. (b) Similar to (a) but for the three models in VAE_$E_3$.

| FactorVAE Metric | | | | | |
|---|---|---|---|---|---|
| Individual Model | | | VAE Ensemble (VAE_E) | | |
| VAE | $0.635\pm0.083$ | | $\gamma=1$ | $\gamma=5$ | $\gamma=10$ |
| $\beta$-VAE ($\beta$=4) | $0.665\pm0.089$ | VAE_$E_2$ | $0.711\pm0.106$ | $0.736\pm0.085$ | $0.741\pm0.086$ |
| FactorVAE ($\gamma$=40) | $0.764\pm0.075$ | VAE_$E_3$ | $0.794\pm0.030$ | $0.792\pm0.075$ | $0.821\pm0.066$ |
| DIP-VAE-I ($\lambda_{od}$=5) | $0.638\pm0.108$ | VAE_$E_4$ | $0.833\pm0.037$ | $0.790\pm0.038$ | $0.800\pm0.078$ |
| DIP-VAE-II ($\lambda_{od}$=5) | $0.676\pm0.122$ | VAE_$E_5$ | $0.828\pm0.016$ | $0.786\pm0.051$ | $0.739\pm0.085$ |
| TC-VAE ($\beta$=4) | $0.808\pm0.079$ | | | | |
| DCI-Disentanglement Metric | | | | | |
| Individual Model | | | VAE Ensemble (VAE_E) | | |
| VAE | $0.143\pm0.033$ | | $\gamma=1$ | $\gamma=5$ | $\gamma=10$ |
| $\beta$-VAE ($\beta$=4) | $0.198\pm0.076$ | VAE_$E_2$ | $0.176\pm0.043$ | $0.243\pm0.029$ | $0.201\pm0.037$ |
| FactorVAE ($\gamma$=40) | $0.253\pm0.072$ | VAE_$E_3$ | $0.214\pm0.064$ | $0.236\pm0.051$ | $0.311\pm0.060$ |
| DIP-VAE-I ($\lambda_{od}$=5) | $0.049\pm0.017$ | VAE_$E_4$ | $0.240\pm0.059$ | $0.223\pm0.045$ | $0.251\pm0.038$ |
| DIP-VAE-II ($\lambda_{od}$=5) | $0.106\pm0.032$ | VAE_$E_5$ | $0.242\pm0.032$ | $0.244\pm0.039$ | $0.196\pm0.050$ |
| TC-VAE ($\beta$=4) | $0.303\pm0.052$ | | | | |

Table 1: Comparison between the proposed VAE ensemble, the original VAE, and the current state-of-the-art disentangled VAE models. We report the mean and standard deviation of the FactorVAE metric and and DCI Disentanglement scores over 10 runs trained on the *dSprite* data.

**Q3:** In Table 1, we compare the disentangled representation performance between the proposed VAE ensemble and the state-of-the-art models. For the VAE ensemble, we report the performance of the first model in the ensemble. We also report the results for the VAE ensemble with different ensemble size and $\gamma$ values. As shown in the table, the VAE ensemble significantly improves the performance over the original VAE model. In many settings, the VAE ensemble achieves similar or better performance over the state-of-the-art models. In Table 2, we evaluate the consistency among the models in the ensemble by reporting the standard deviation of the evaluation metrics using different models in the same ensemble. The small values confirm that different models in the ensemble learn similar latent representations. Furthermore, Table 1 shows the joint effect of ensemble size and $\gamma$ setting. When $\gamma = 1$, the performance of VAE ensemble increases as the ensemble size increases, indicated by the higher mean and smaller variance of both the FactorVAE and DCI Disentanglement metrics. This behavior is consistent with the characteristic of ensemble learning where the increase in performance becomes smaller as the size of ensemble increases. However, as $\gamma$ increases, having larger ensemble size can reduce the performance. We believe this

|  | $\gamma$ | VAE_$E_2$ | VAE_$E_3$ | VAE_$E_4$ | VAE_$E_5$ |
|---|---|---|---|---|---|
| FactorVAE Metric | ($\gamma$=1) | 0.0019 | 0.0090 | 0.0048 | 0.0081 |
|  | ($\gamma$=5) | 0.0058 | 0.0064 | 0.0089 | 0.0163 |
|  | ($\gamma$=10) | 0.0060 | 0.0046 | 0.0147 | 0.0139 |
| DCI-Disent Metric | ($\gamma$=1) | 0.0024 | 0.0026 | 0.0028 | 0.0036 |
|  | ($\gamma$=5) | 0.0037 | 0.0054 | 0.0049 | 0.0040 |
|  | ($\gamma$=10) | 0.0013 | 0.0024 | 0.0041 | 0.0042 |

Table 2: The comparison between individual models in the same ensemble. We report the average of the standard deviation of the metrics by individual models in the ensemble across 10 runs.

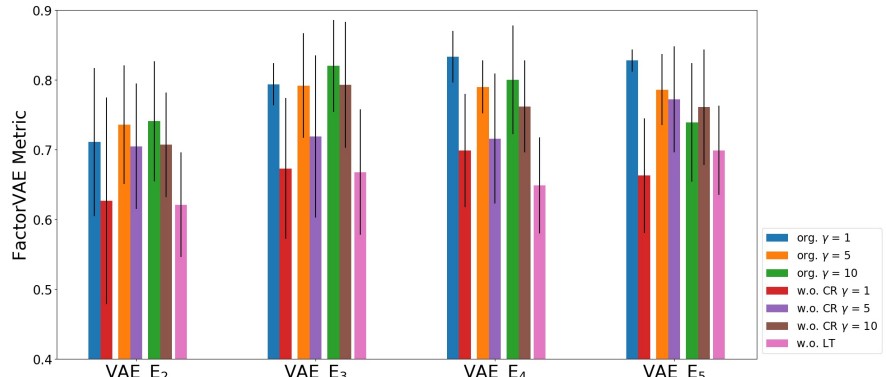

Figure 4: Ablation study to understand the effect of cross-model reconstruction and linear transformation in the VAE ensemble objective using the FactorVAE metric. (*w.o. CR* - without cross-model reconstruction loss; *w.o. LT* - without linear transformation loss; *org* - original VAE ensemble loss)

is due to the increased difficulty of balancing between the cross-model and within model objectives of VAE ensemble for larger ensembles. The reduced alignment of the latent representations among different models can also be seen in Table 2 where difference in the performance among individual models in the ensemble increases as ensemble size increases.

**Q4:** We conduct the ablation study to further understand the effect of the linear transformation loss and the cross-model reconstruction loss in the VAE ensemble objective. As shown in Fig. 4, removing either component leads to a lower FactorVAE metric for the VAE ensemble. Without the linear transformation loss, the performance of VAE ensemble decreases significantly across different ensemble sizes. Without the cross-model reconstruction loss, the performance of VAE ensemble also decreases but the gap becomes smaller as $\gamma$ increases. This matches the discussion in Section 3 that higher $\gamma$ forces closer mapping between the encoders and reduce the cross-model reconstruction error of the decoders. However, this also reduces the effect of cross-model reconstruction as discussed in Section 4.2. A similar result is also found for the DCI Disentanglement metric as shown in Fig. 8 in Appendix E. Overall, the results from the ablation study confirms the importance of both the linear transformation loss and the cross-model reconstruction loss in the VAE ensemble objective.

## 6 CONCLUSION

In this study, we propose a simple yet effective VAE ensemble framework consisting of multiple original VAEs to learn disentangled representation. The individual models in the ensemble are connected through linear layers that regularize both encoders and decoders to align the latent representations to be similar up to a linear transformation. We show in theory that the regularization by the VAE ensemble forces the linear transformations to be trivial transformations and show improved performance on the unsupervised disentangled representation learning. The theoretical discussion in Section 4 is based on the original VAE objective, and our experiments also focus on the ensemble with original VAE. We believe such framework can be extended to other disentangled VAE models, or even a mixture of different VAE models, as long as the regularization by the ensemble does not conflict with the augmented objective of these models.

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

## A    TECHNICAL LEMMAS

In this section, we give the lemmas used in the theoretical discussion in Section 4.

**Lemma 1.** *(Jensen's inequality) If $g(x)$ is a convex transformation on $x$, then this convex transformation of a mean $g[\mathbb{E}(x)]$ is less than or equal to the mean of the convex transformed value $\mathbb{E}[g(x)]$; it is a simple corollary that the opposite is true of concave transformations.*

**Lemma 2.** *(AM-GM inequality) As an extension of Jensen's inequality, given a list of non-negative real numbers $x_1, x_2, \ldots, x_n$, the arithmetic mean of this list $\frac{1}{n} \sum_{i=1}^{n} x_i$ is greater than or equal to the geometric mean of the same list $(\prod_{i=1}^{n} x_i)^{\frac{1}{n}}$; and further, the equality holds if and only if $x_1 = x_2 = \cdots = x_n$.*

**Lemma 3.** *(Hadamard's inequality). if $M$ is the matrix having columns $c_i$, then $|\det(M)| \leq \prod_{i=1}^{n} \|c_i\|$; and the equality in Hadamard's inequality is achieved if and only if the vectors are orthogonal.*

## B    DISTANCE TO ORTHOGONALITY (DTO)

In this section, we introduce the detail of *Distance to Orthogonality* (DtO) that is used in our experiment to check if the linear transformations in the VAE ensemble approach trivial transformations. This measurement is also used in (Rolinek et al., 2019) for a similar purpose. DtO is the Frobenius norm of the difference between a square matrix $M$ and its closest signed permutation matrix $P(M)$. Finding $P(M)$ can be formulated as a mixed-integer linear programming (MILP) problem as following:

$$
\begin{aligned}
\min_{P} \quad & \sum_{i,j} |M_{i,j} - P(M)_{i,j}| \\
s.t. \quad & P(M)_{i,j} \in \{-1, 0, 1\}, \quad \forall (i,j) \\
& \sum_{i} |P_{i,j}| = 1, \quad\quad\quad \forall j \\
& \sum_{j} |P_{i,j}| = 1, \quad\quad\quad \forall i
\end{aligned}
\tag{8}
$$

By introducing new variables $P_{i,j}^{+}, P_{i,j}^{-} \in \{0, 1\}$ and $D_{i,j} = |M_{i,j} - P(M)_{i,j}|$, we can reformulate the above optimization problem as:

$$
\begin{aligned}
\min_{P} \quad & \sum_{i,j} D_{i,j} \\
s.t. \quad & (P_{i,j}^{+} - P_{i,j}^{-}) - M_{i,j} \leq D_{i,j}, \quad \forall (i,j) \\
& M_{i,j} - (P_{i,j}^{+} - P_{i,j}^{-}) \leq D_{i,j}, \quad \forall (i,j) \\
& \sum_{i} (P_{i,j}^{+} + P_{i,j}^{-}) = 1, \quad\quad\quad \forall j \\
& \sum_{j} (P_{i,j}^{+} + P_{i,j}^{-}) = 1, \quad\quad\quad \forall i
\end{aligned}
\tag{9}
$$

Using this optimization formulation, DoT of a given matrix $M \in \mathbb{R}^{n \times n}$ is defined as:

$$
DoT = \frac{1}{n^2} \sum_{i,j} |M_{i,j} - P(M)_{i,j}|
\tag{10}
$$

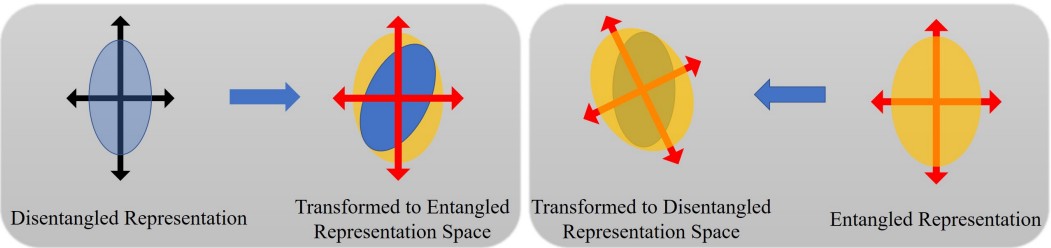

Figure 5: Geometric interpretation of the cross-model reconstruction between a disentangled representation space and an entangled representation space.

## C  GEOMETRIC INTERPRETATION OF THE EFFECT OF CROSS-MODEL RECONSTRUCTION

Given a disentangled and entangled latent representation space, Fig. 5 illustrates the effect of the cross-model reconstruction by VAE ensemble. The left part shows the orthogonal transformation from a disentangled representation to an entangled space, and the right part shows the transformation in the opposite direction. As shown in the figure, the orthogonal transformation from the disentangled representation to the entangled space does not introduce larger variance than the entangled representation. Hence, we can expect similar cross-model reconstruction and within model reconstruction. However, the transformation from the entangled representation to the disentangled space introduces larger variance (yellow shaded area over blue area on the right) than the disentangled representation. This leads to larger cross-model reconstruction by the disentangled model.

## D  MODEL ARCHITECTURE AND TRAINING DETAILS

We conducted our experiments, including training and evaluating the current state-of-the-art disentanglement models as well as evaluating the proposed VAE ensembles, using the `disentanglement_lib` [1] open-source library (Locatello et al., 2018).

Table 3 shows the encoder and the decoder architecture of the VAE model used in our experiments. This architecture is the same as the one used in the original $\beta$-VAE Higgins et al. (2017).

| Encoder | Decoder |
|---|---|
| Input 64×64 binary/RGB image | Input $\mathbb{R}^d$ |
| 4×4 conv, 32 ReLu, stride 2, pad 1 | FC $d$×256, ReLu |
| 4×4 conv, 32 ReLu, stride 2, pad 1 | 4×4 upconv, 64 ReLu, stride 1 |
| 4×4 conv, 64 ReLu, stride 2, pad 1 | 4×4 conv, 64 ReLu, stride 2, pad 1 |
| 4×4 conv, 64 ReLu, stride 2, pad 1 | 4×4 conv, 32 ReLu, stride 2, pad 1 |
| 4×4 conv, 256 ReLu, stride 1 | 4×4 conv, 32 ReLu, stride 2, pad 1 |
| FC 256 × (2×$d$) | 4×4 conv, $nc$ , stride 2, pad 1 |

Table 3: Encoder and Decoder architecture, $d$: dimension of the latent representation; $nc$: number of input image channel (For *dSprites* dataset $nc = 1$, for *CelebA* dataset $nc = 3$).

Table 4 shows the hyperparameters setting used throughout the experiments. These parameters are fixed for all the experiments.

## E  ADDITIONAL EXPERIMENTAL RESULTS

In this section, we present the additional results including the DtO and "polarized regime" analysis on the models trained on the *CelebA* dataset similar to the ones conducted on *dSprite* dataset in Section 5; the ablation results with DCI-Disentanglement metric and the DtO estimation; and the

---

[1]https://github.com/google-research/disentanglement_lib

| Parameter | value |
|---|---|
| Batch size | 64 |
| Latent dimension | 10 |
| Optimizer | Adam |
| Adam: beta1 | 0.9 |
| Adam: beta2 | 0.999 |
| Learning rate | 1e-4 |

Table 4: Hyperparameters setting.

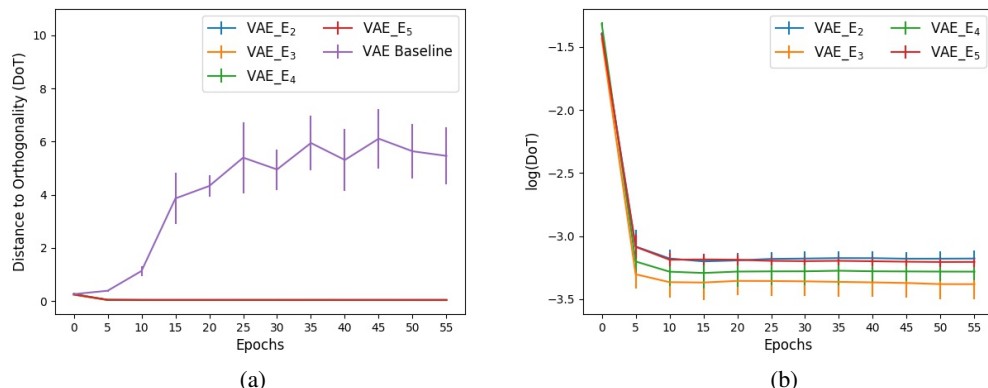

(a)                                                   (b)

Figure 6: Characteristics of the linear transform between latent representations. The latent dimension is set to 10 and the results are over 10 runs of training on the *dSprite* dataset. (a) Comparing the DtO of linear transformations in the VAE ensemble ($\gamma$=10) and the one between original VAEs. (b) VAE ensemble ($\gamma$=10) with different ensemble size all achieve small DtO of the linear transformations between the models.

latent traversal of the trained model on both *dSprite* and *CelebA* dataset along with further discuss the effect of VAE ensemble on the latent representation.

### E.1   CHARACTERIZATION OF THE LINEAR TRANSFORMATION IN VAE ENSEMBLE

In Figure 6, we show the DtO estimation of the linear transformations in the ensemble during training for the *dSprite* dataset. We report the mean and standard deviation of DtO across all linear transformations over 10 different runs. Furthermore, we compare these results with a VAE baseline where DtO is estimated for the linear transformations between original VAEs. Specifically, we train ten different VAEs separately and estimate the DtO of the pairwise linear transformation among these models during training. Similarly we report the mean and standard deviation of these DtO estimations. As seen in the figure, the VAE ensemble models with different ensemble size all approach to trivial transformations between the individual models, while the original VAEs do not have such property. A similar result is also found in models trained for *CelebA* dataset. Similar to the results in Figure 6, we observe decreased DtO of the linear transformations in the VAE ensemble during training.

We also compare models trained with different latent dimension size. We observe decreased DtO as the latent dimension of the model increases in Figure 7. This is because, as discussed in the main paper, the VAE ensemble encourages more "active" latent variables. Models with higher latent dimension likely to learn a decomposition of generative factors. As a result, the alignment of the latent variable between different models are easier thus the linear transformations between the latent representations is closer to the trivial transformation. On the contrary when there are less latent variables in the model than the generative factors, some of the latent variables will capture more than a single generative factor. As a result, the one-to-one mapping between the latent variables of different models will not lead to a trivial transformation.

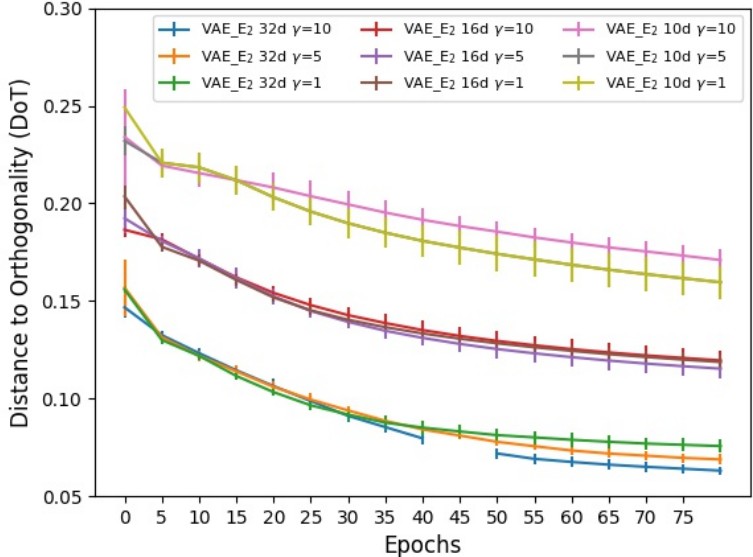

Figure 7: Distance to Orthogonality (DtO) measurement of the linear transforms between latent representations in VAE_$E_2$ during training on the *CelebA* dataset. We also compare models with different latent dimensions of 10, 16 and 32 and the results are averaged over 5 runs. In the figure legend, we use "VAE_$E_i$ $n$d $\gamma = g$" to represent VAE Ensemble (VAE_E) model with $i$ individual VAE models, $n$ latent dimensions and $\gamma$ value equal to $g$.

In Figure 9 and Figure 10, we show the "polarized regime" estimation for models in VAE_$E_2$ and VAE_$E_3$ trained for *CelebA* datasets, respectively. Similar to the results in Figure 3, individual models in the VAE ensemble tend to have similar 'polarized regime", and higher $\gamma$ enforces the "polarized regime" by separating "passive" latent variables from the "active" ones. When we compare between VAE_$E_2$ and VAE_$E_3$, we observe increased "active" latent variables similar to the result on *dSprite* dataset in Section 5. More importantly, as discussed earlier, latent variables in a model with limited latent dimensions need to capture more than a single generative factor, especially for a complicated real-world dataset such as *CelebA*. This makes the linear transformation between the latent representations less trivial. As the latent dimension size grows, such constraint is relaxed and the linear transformations are closer to trivial.

These additional results confirm the conclusion in Section 5: (1) as the ensemble size increases, DtO increases due to the difficulty of aligning the latent representations among different models; (2) as the model latent dimension increases, DtO decreases due to the increased model capacity, and encourages the one-to-one mapping between latent variables in different models; (3) hyperparameter $\gamma$ does not affect DtO significantly, but plays an important role on separating "active" and "passive" latent variables, especially when the latent dimension is large enough.

Furthermore, we believe the DtO measurement of the linear transformation in VAE ensemble could be a useful indicator for latent dimension size. As shown in Figure 7 and Figure 9, when the latent dimension is sufficient for a given dataset, the DtO of the linear transformation is small and some latent variables are pushed to "passive" mode.

### E.2   ABLATION STUDY

Similar to the ablation result shown in Section 5, here we show the same ablation study using the DCI Disentanglement metric in Fig. 8(a) as well as the DtO measurement 8(b). Similar as the results of the FactorVAE metric in Fig. 4, removing either component leads to a lower DCI Disentanglement metric for the VAE ensemble. Without the linear transformation loss, the performance of VAE ensemble decreases significantly across different ensemble sizes. Fig. 8(b) shows that for VAE

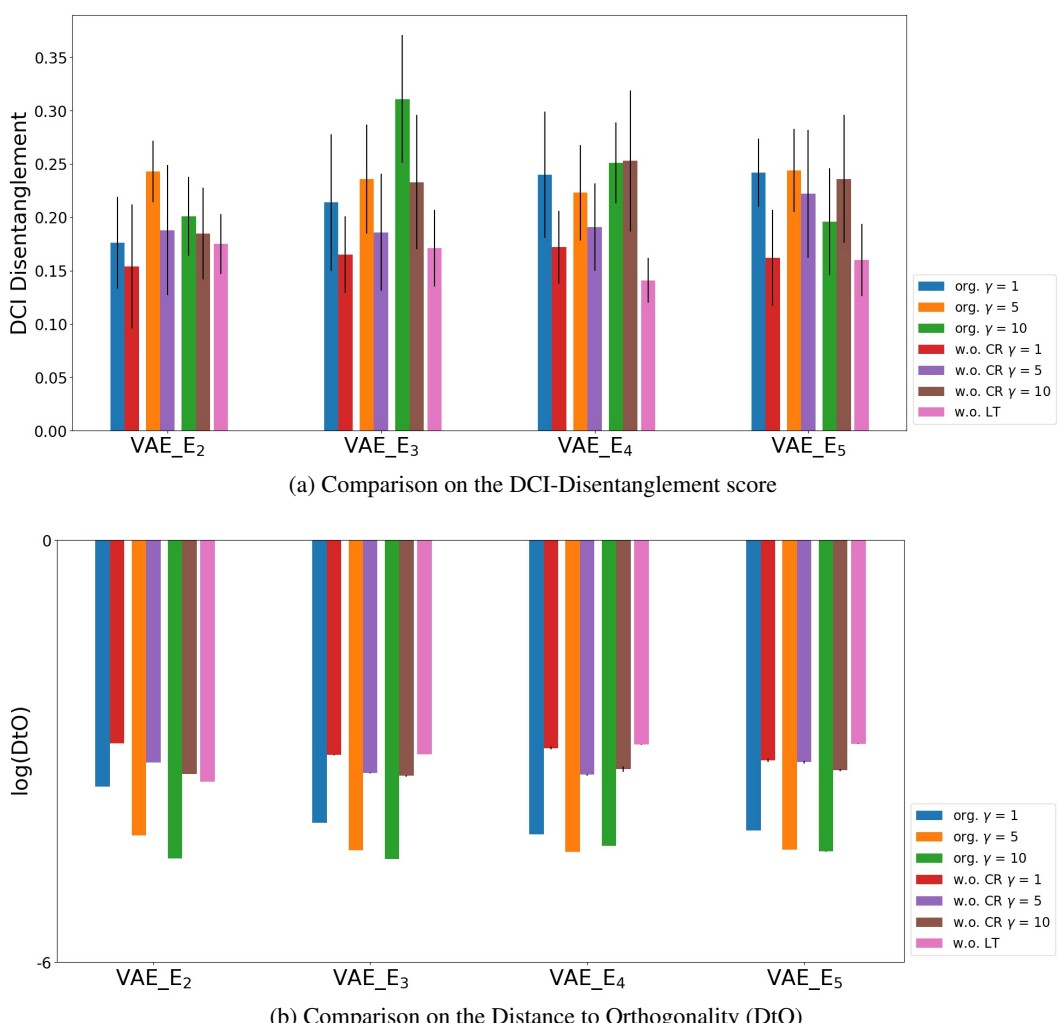

(a) Comparison on the DCI-Disentanglement score

(b) Comparison on the Distance to Orthogonality (DtO)

Figure 8: Ablation study to understand the effect of cross-model reconstruction and linear transformation in the VAE ensemble objective using the DCI Disentanglement metric and DtO. (*w.o. CR* - without cross-model reconstruction loss; *w.o. LT* - without linear transformation loss; *org* - original VAE ensemble loss)

ensemble without the cross-model reconstruction, the linear transformations among models are close to a trivial transformation (signed permutation). This implies the orthonormal transformation of the linear transformations. This result further supports our intuitive justification in Appendix C that the cross-model objective encourages entangled models to align to disentangled models. Indeed, we see that adding the cross-model reconstruction can further reduce the DtO of the linear transformations among the models in the ensemble.

### E.3 LATENT TRAVERSAL

In this section we show the latent traversal of models trained on both *dSprites* and *CelebA* datasets. For a fixed input image, to extract the latent traversal we change the value of a single latent variable $z_i$ in the corresponding encoding, and observe the generated output image to understand the effect of $z_i$. The range of the value are usually chosen to be from -3 to 3 due to the standard Gaussian prior.

In Figure 11, we show the latent traversal for both VAE_$E_2$ and a single VAE model with 10 latent dimensions trained on *dSprites* dataset. Three images as shown in the last column of each block

are used as input. Both models are able to capture certain generative factors of the data including position, shape, rotation and scale. In Section 5, we argue that the representation by VAE ensemble encourages more "active" latent variables, thus can capture a decomposition of the ground truth generative factors. Especially from the "polarized regime" estimation in Figure 3, we observe that some latent variables in the VAE ensemble are in-between "active" and "passive" modes. This suggests that the VAE ensemble model generates input-dependent factors based on the input complexity. In Figure 11, we observe this behavior highlighted with color boxes. The traversal on the second latent variable $z_2$ shows that an ellipse shape does not lead to an "active" latent variable. However, both heart and square shape lead to an "active" latent variable that changes the output. In contrast, the single VAE model does not have such behavior where the "active" modes are consistent across different input data.

In Figure 12 and Figure 13, we show the latent traversal for both VAE_$E_2$ and a single VAE model with 16 latent dimensions trained on *CelebA* dataset, respectively. In this real-world dataset, the generative factors are unknown. We observe different factors including background, azimuth, gender, hair style being captured by both models. Similar as before, the single VAE model maintains similar "active" mode for all latent variables where similar traversal patterns are observed for both input images. However, VAE_$E_2$ shows semantically consistent but input-dependent "active" mode. This is translated into different traversal effects and more realistic and sharper images by VAE_$E_2$, especially for the first input image that is less common in the dataset. We believe this is important towards a meaningful compositional latent representation learning.

Overall, the latent traversal results in this section confirm the findings on the inner working of the VAE ensemble shown in the previous section as well as the discussion in Section 5.

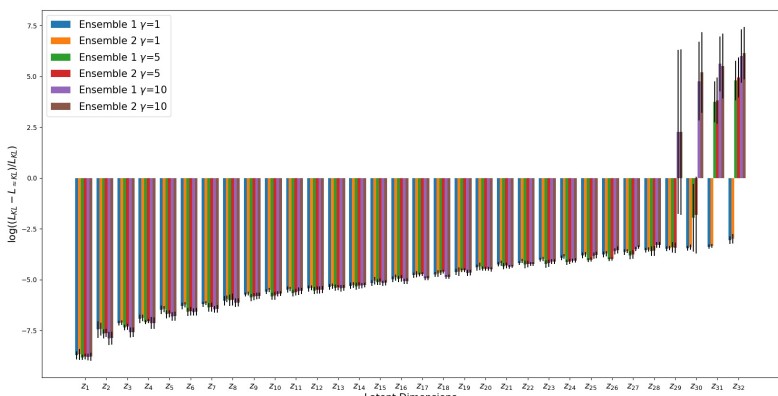

(a) VAE_$E_2$ with latent dimension of 32, CelebA data.

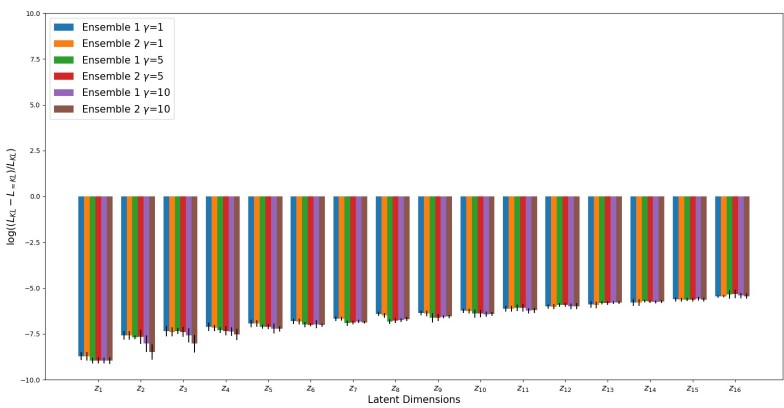

(b) VAE_$E_2$ with latent dimension of 16, CelebA data.

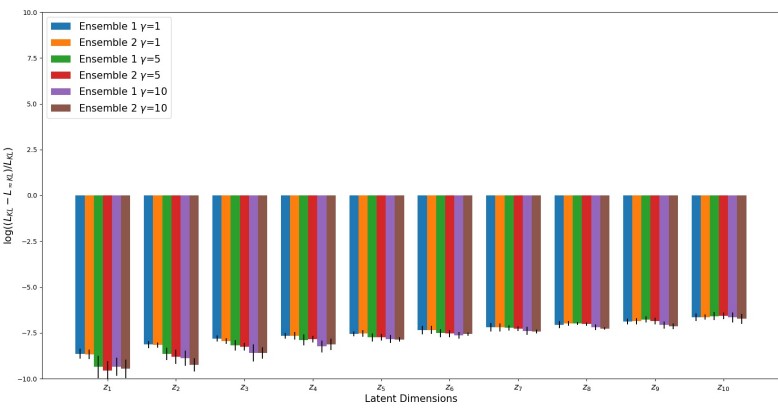

(c) VAE_$E_2$ with latent dimension of 10, CelebA data.

Figure 9: The "polarized regime" comparison between models in VAE_$E_2$. The results are over 5 runs of training on the *CelebA* dataset.

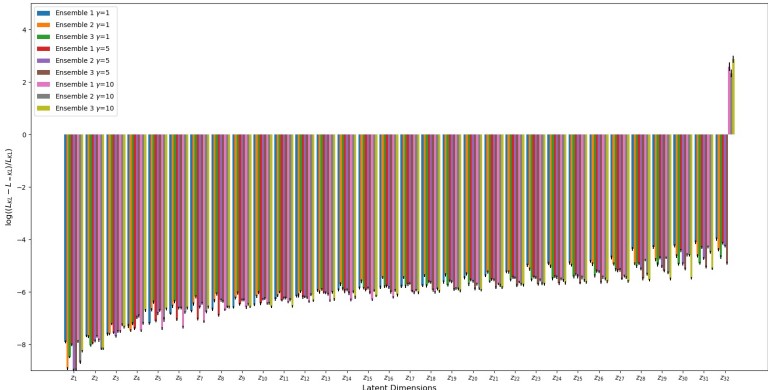

(a) VAE_$E_3$ with latent dimension of 32, CelebA data.

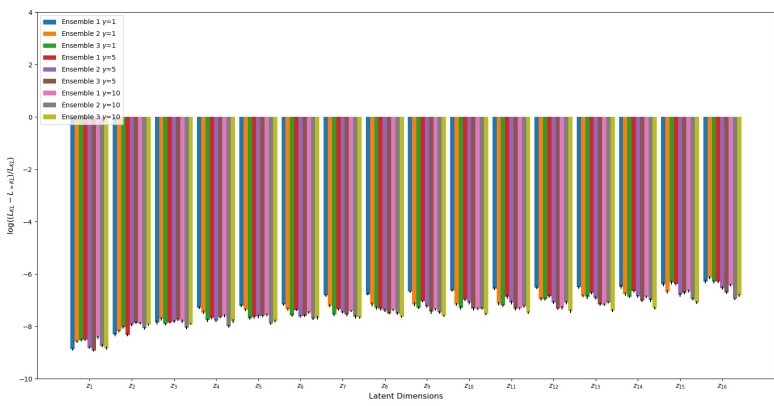

(b) VAE_$E_3$ with latent dimension of 16, CelebA data.

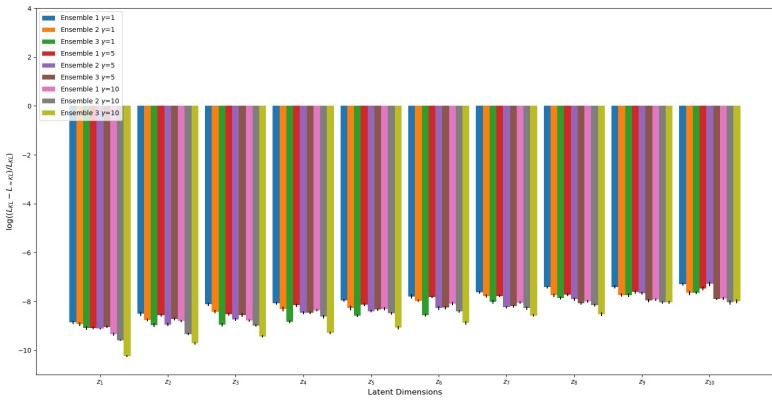

(c) VAE_$E_3$ with latent dimension of 10, CelebA data.

Figure 10: The "polarized regime" comparison between models in the VAE_$E_3$ on the *CelebA* dataset.

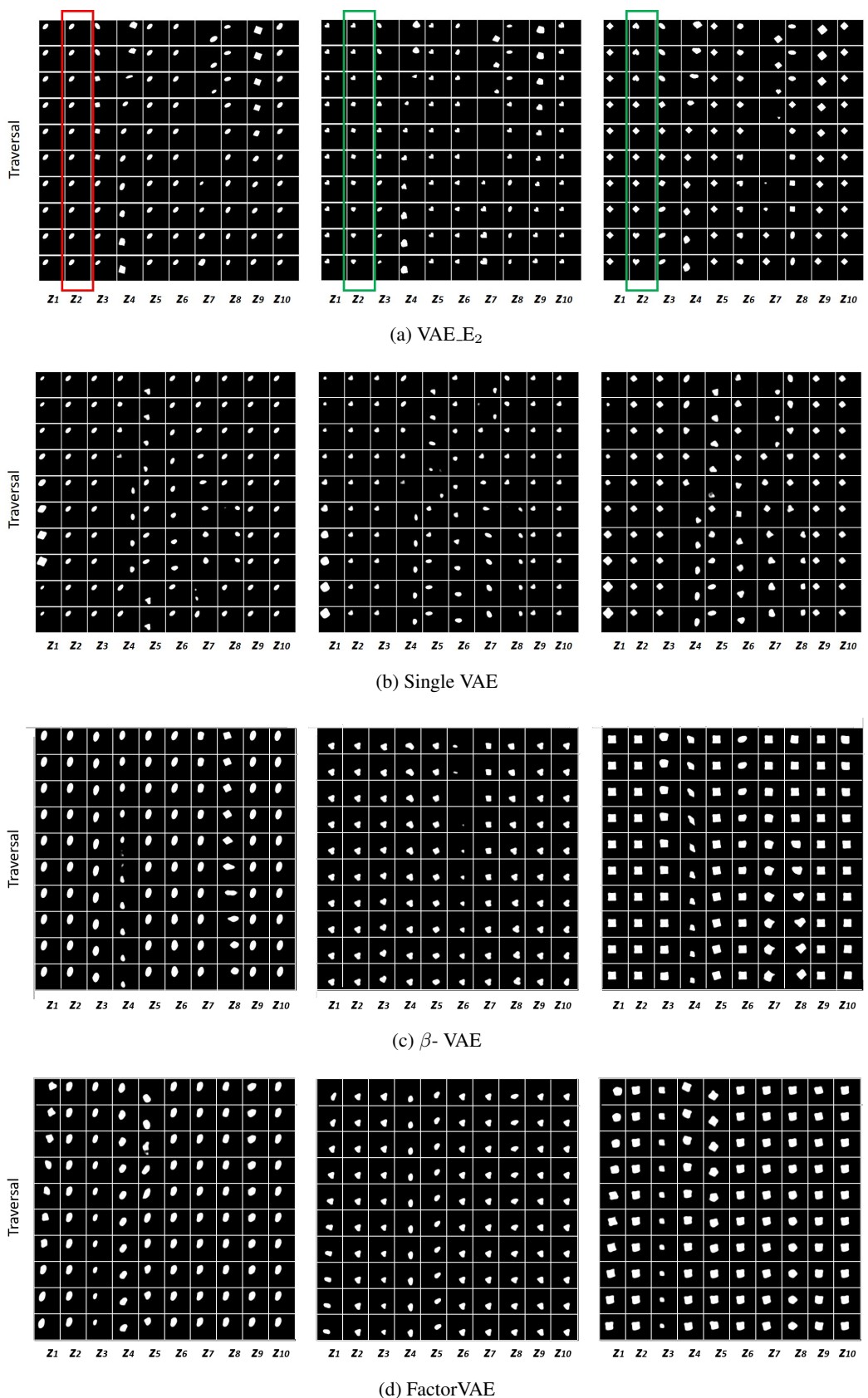

Figure 11: Latent traversal on three different input images using VAE_$E_2$, a single VAE and the state-of-the-art VAE models with 10 dimensional latent representation. The three input images are ellipse, heart and square shapes as shown in the last column.

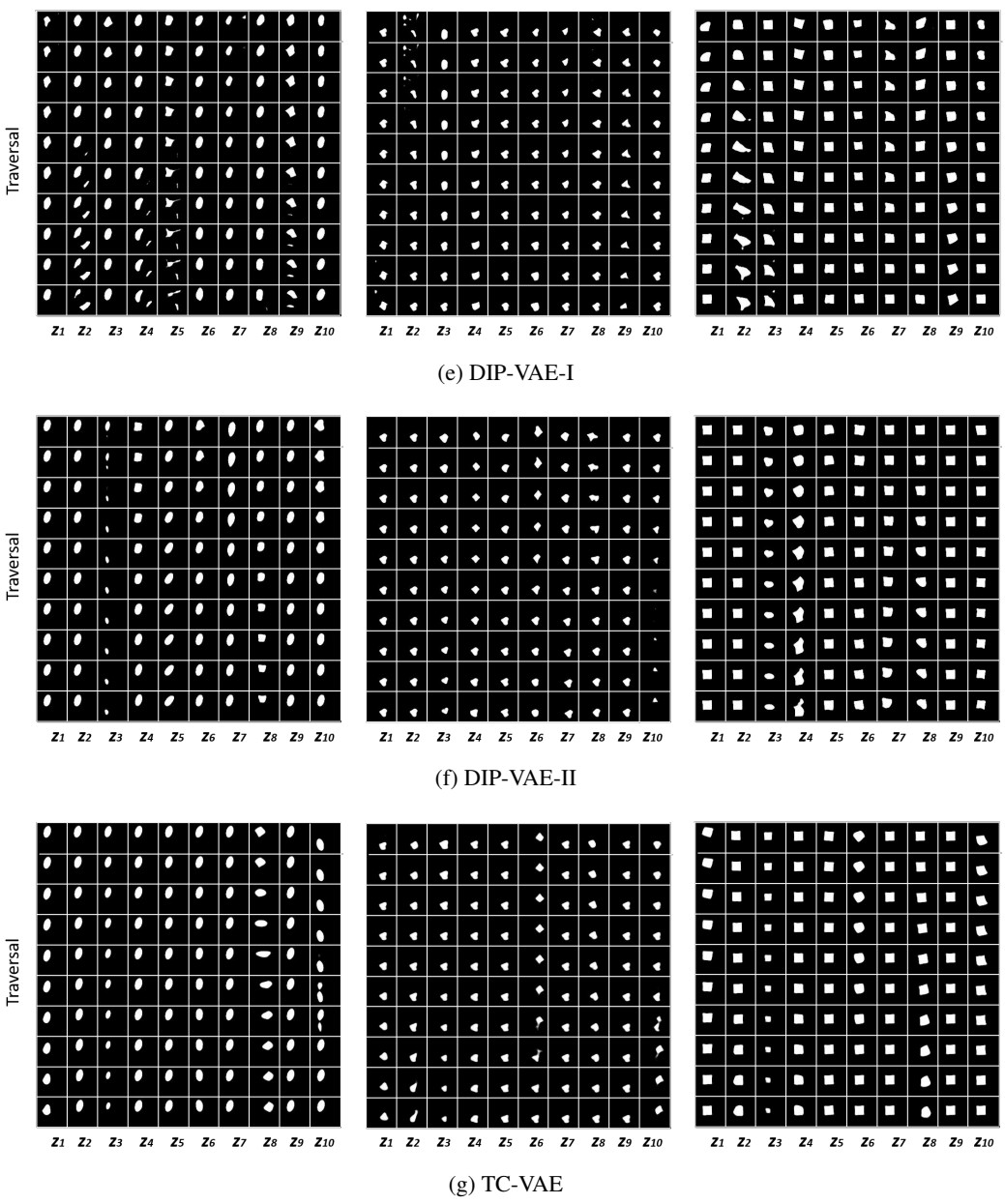

Figure 11: (cont.) Latent traversal on three different input images using VAE_E$_2$, a single VAE and the state-of-the-art VAE models with 10 dimensional latent representation. The three input images are ellipse, heart and square shapes as shown in the last column.

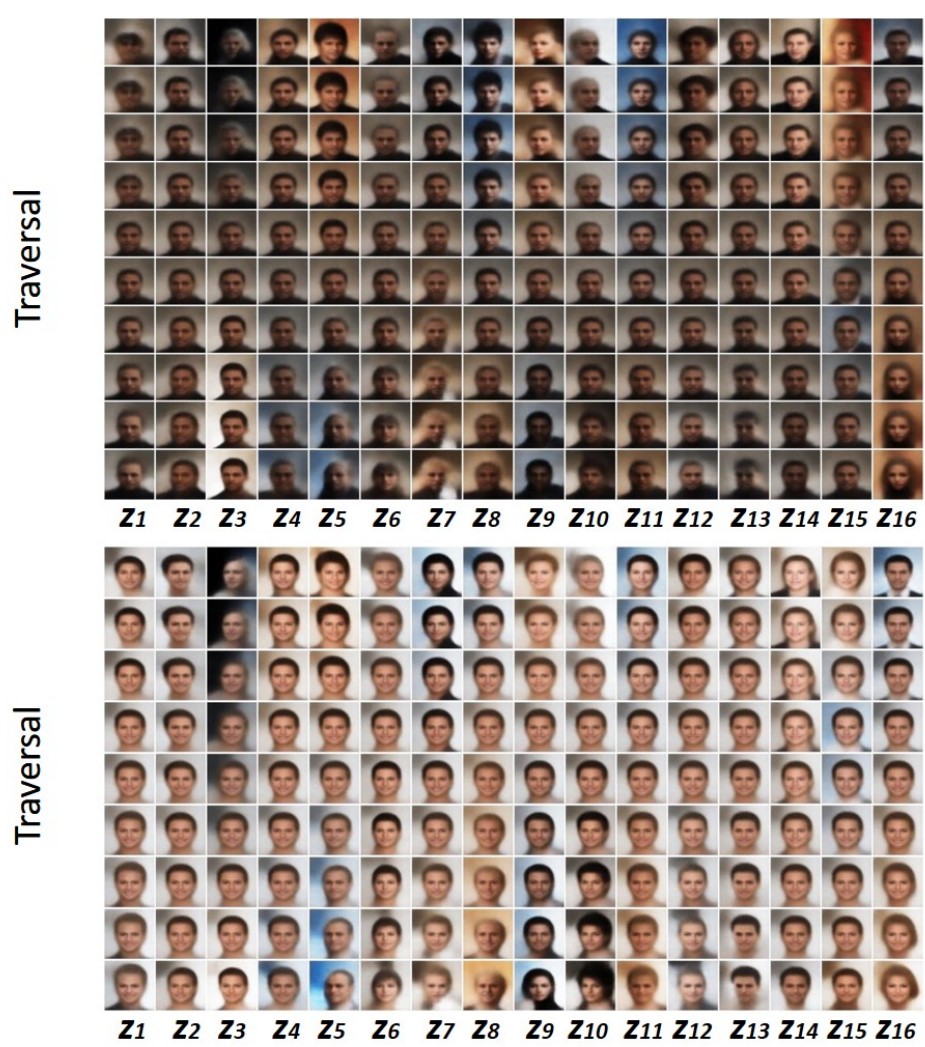

Figure 12: Latent traversal on two different input images of *CelebA* dataset using VAE_E$_2$ with latent dimension of 16.

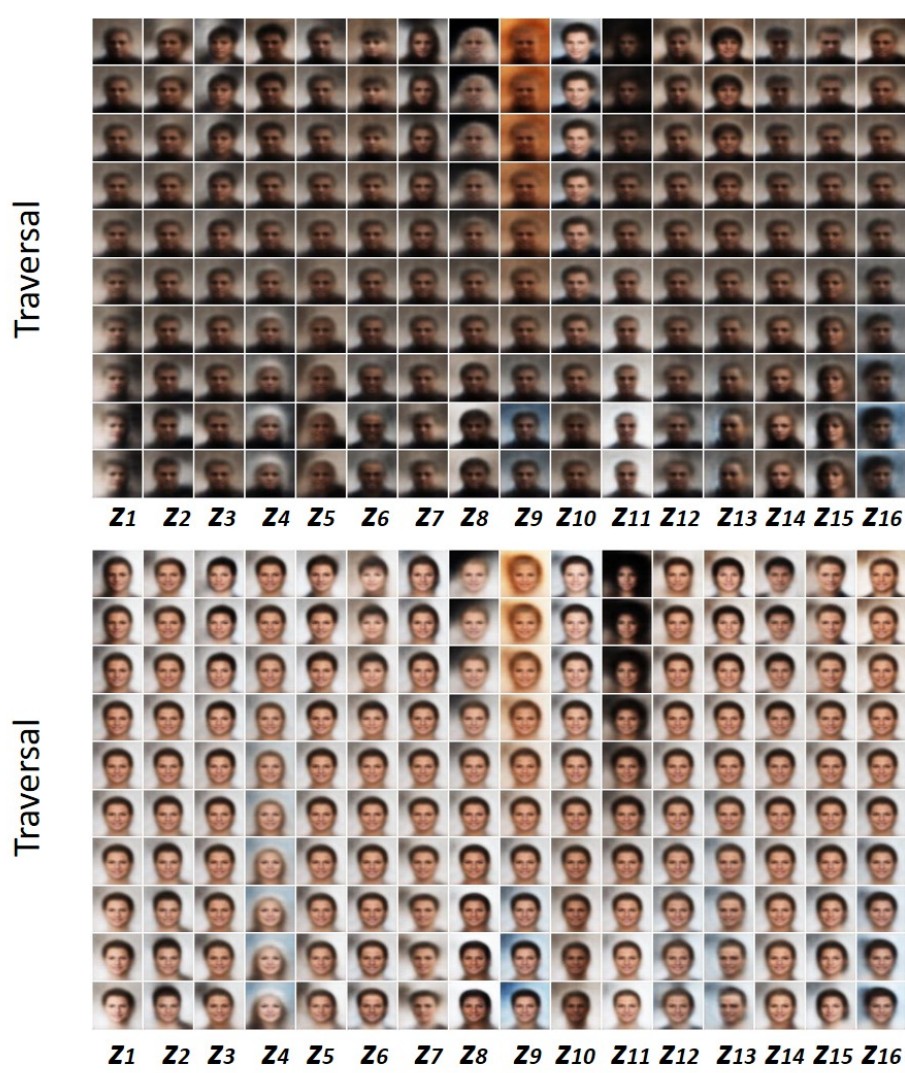

Figure 13: Latent traversal on two different input images of *CelebA* dataset using a single VAE with latent dimension of 16.

