# OpenReview forum: "Improving the Unsupervised Disentangled Representation Learning with VAE Ensemble"
_ICLR.cc/2021/Conference — Reject_

### Official Review · AnonReviewer1 · 2020-10-24
**Recommend to Reject**

**Rating:** 3
**Confidence:** 4

**Review:**

# Summary

The authors introduce a novel VAE-based approach for unsupervised learning of disentangled representations of image data.  The approach trains an ensemble of VAEs along with pair-wise linear transformations between their latent spaces.  The objective includes the ELBO objectives for each VAE as well as two additional pressures:  (i) An L2 similarity objective that pressures samples from each VAE latent space to match under linear transformations samples from the other VAE latent spaces, and (ii) A cross-model decoding objective that encourages decoding accuracy of the linearly transformed latent samples.  The authors provide a theoretical argument that the linear transformations should learn to be orthogonal, and show some experimental results indicating that their model performs well compared to baselines when evaluated with an established disentangling metric.

# Pros

* The theoretical analysis in section 4.1 is clear and provides good mathematical intuition for the authors’ results.
* The introduction and related work sections are clear and include a thorough set of references.


# Cons:

* The authors’ baseline results give unexpectedly low metric scores.  The authors report FactorVAE metric values of 0.665 for beta-VAE and 0.764 for FactorVAE on the dSprites dataset.  However, the values reported in the FactorVAE paper (and corroborated by others) on the same dataset are significantly higher.  This makes me suspicious that something went wrong with the authors’ training --- perhaps they didn’t train those baseline models to completion or something else went wrong.  Having baseline results that are inconsistent with the existing literature makes me uneasy.
* The traversals in Figure 8 from the authors’ model are much less disentangled than other models in the literature.  For example, they are much less disentangled than the traversals shown in the beta-VAE paper and the FactorVAE paper on the same dataset.  Thus from these traversals, it seems that the authors’ model is performing worse than existing models in the literature (the authors’ metrics indicate the opposite, but as mentioned above I’m uncertain about the validity of those metric results).  Figure 3-A also suggests that the authors’ model is using too many informative latents, i.e. not disentangling well.
* I am not convinced by the authors’ intuitive justification in lines 216-225 (and appendix C) that the cross-model objective encourages entangled models to align to disentangled models.  Specifically, in that argument the authors seem to assume that orthogonal linear transformations are orthonormal.  However, there is nothing to enforce normality of the transformations in the model, hence the cross-model encoding variance from an entangled to a disentangled model could be quite small.
* The purpose of the cross-model reconstructions is not clear, particularly given that I’m not convinced by the authors’ intuitive justification of them.  The L2 regularization between the transformed encodings should pressure the cross-model reconstructions to be good, so I do not see the reason to include them in the model objective.  It would be good if the authors could do an ablation study without the cross-model reconstructions.
* The authors do not mention the computational complexity of their model, yet computational complexity seems to be a significant drawback of it.  Ensemble training is very computationally expensive, so the authors should include some discussion about it as well as runtimes and memory requirements for their model.  Furthermore, with the cross-model reconstructions the computational complexity of the authors’ model scales with the square of the number of ensemble elements, which is quite a steep scaling.
* The authors only compare to a couple (relatively old) baselines, betaVAE and FactorVAE, which are no longer state-of-the-art.  However, more recently a number of other VAE models have been published that perform better.  In order to support their claims about state-of-the-art performance, the authors should compare to newer baselines.  Here are a few examples:
DIP-VAE  (Variational inference of disentangled latent concepts from unlabeled observations.  Kumar et al., 2017)
TCVAE (Isolating sources of disentanglement in variational autoencoders. Chen et al., 2018)
Spatial Broadcast VAE  (Spatial Broadcast Decoder: A Simple Architecture for Learning Disentangled Representations in VAEs.  Watters et al., 2019)
* The authors also don’t include many metrics or datasets.  dSprites and CelebA were used in the original betaVAE paper, but more recently it has become the norm to test on a larger set of datasets and with a number of different metrics to convincingly show disentangling.  By the way, a number of models, datasets, and metrics have been open-sourced in DistLib (https://github.com/google-research/disentanglement_lib), which may be useful for comparing to more models with more metrics on more datasets.

# Summary

I do not recommend accepting this paper.  Baseline results are inconsistent with prior work, the model seems to disentangle less well than existing methods, and the authors don’t do ablation experiments to justify the high computational complexity of the model.

---

> ### Author Response · Authors · 2020-11-18
> **Response to AnonReviewer1**
>
> Thank the reviewer for raising the concerns! Below we answer each question from the reviewer.
>
> •	As the reviewer suggested, we used the open-source distentanglement_lib to conduct our study, including the baseline results on beta-VAE and FactorVAE. We added this important implement detail in Appendix D of the paper. Our baseline results are consistent with the recent large-scale experiment results in (Locatello et al., 2018) and (Duan et al. 2019). Notice that the disentanglement_lib is published with (Locatello et al., 2018). The discrepancy between these results and the ones reported in the original FactorVAE paper is due to the parameter setting for estimating the FactorVAE metric. It involves training a classifier on a training data and reporting the classifier performance on an evaluation dataset. The parameters during this process include the size of training and evaluation, the batch size of training the classifier, the number of data points to estimate the variance of each latent dimension and the threshold for pruning the latent variables. We report the results of all the experiments following the same parameter settings as the two studies mentioned above.
>
> •	The representation by VAE ensemble encourages more “active” latent variables, thus can capture a decomposition of the ground truth generative factors. Especially from the “polarized regime” estimation in Figure 3, we observe that some latent variables in the VAE ensemble are in-between “active” and “passive” modes. This suggests that the VAE ensemble model generates input-dependent factors based on the input complexity. The traversals in Figure 11 (original Figure 8) shows that an ellipse shape does not lead to an “active” latent variable.  However, both heart and square shape lead to an “active” mode of the second latent variable that changes the output.  In contrast, existing SOTA approaches do not have such behavior where the “active” modes are consistent across different input data. We believe a latent representation that preserve the compositional property at the core is what makes disentangled representations useful. This is verified with the quantitative comparison in Table 1 of the updated paper.
>
> •	Our intuitive justification on the cross-model objective is built on the theoretical discussion in Section 4.1  where we show that the linear transformations among models in the VAE ensemble tend to converge to a trivial transformation (signed permutation). This implies the linear transformations to be close to orthonormal transformation. To verify the effect of the cross-model reconstruction, as the reviewer suggested, we conducted the ablation study where we remove the cross-model reconstruction as well as the linear transformation separately. The results are updated in Figure 4 and 8 in the paper. Without the linear transformation loss, the performance of VAE ensemble decreases significantly across different ensemble sizes. Without the cross-model reconstruction loss, the performance of VAE ensemble also decreases but the gap becomes smaller as the hyperparmeter gamma increases. When gamma value is high, the behavior of the VAE ensemble matches the reviewer’s conjecture that the linear transformation loss would force closer mapping between the encoders and reduce the cross-model reconstruction error of the decoders. However, the cross-model reconstruction is important when gamma is low, where the optimal performance of VAE ensemble is achieved.
>
> •	The reviewer raised a valid point on the increased computation complexity of the proposed ensemble model. Since AnonReviewer4 raised the same question (the first question). Due to response space limitation, we kindly ask the reviewer to refer to the response above.
>
> •	We disagree with the reviewer’s comment that betaVAE and FactorVAE are no longer state-of-the-art. The large-scale experiments in (Locatello et al., 2018) and (Duan et al. 2019) still show comparable performance by these two models comparing with the models mentioned by the reviewer. In order to demonstrate the effectiveness of the proposed VAE ensemble, as the reviewer suggested, we extend our experiments to include more SOTA models including TC-VAE and two versions of DIP-VAE.  We also use two widely used supervised metrics including FactorVAE metric and DCI Disentanglement scores (Eastwood & Williams, 2018) as the quantitative measurements. These metrics are shown to correlate with other common supervised metrics (Locatello et al., 2018).  For example, FactorVAE metric and β-VAE metric (Higgins et al., 2017) capture similar notions, while DCI Disentanglement and Mutual Information Gap (MIG) capture similar notions.  In addition, DCI Disentanglement is closely related to the unsupervised model selection method UDR (Duan et al. 2019). Figure 2, Table 1 and 2 and their corresponding discussions are updated in the “Experiment” section of the paper to reflect these changes.

---

> > ### Author Response · Authors · 2020-11-24
> > **Further clarification on baseline models and visualization, and discussion on additional ablation results**
> >
> > Regarding the baseline results, since we did not receive any feedback from the reviewer, we want to further clarify that the results on the baseline models are from re-running the model training and evaluation provided by the disentanglement_lib library (with different random seeds and model initializations). In fact, we compared our results with the pre-trained models published with the disentanlgement_lib and observe similar results (for example, the randomly chosen 10 pre-trained FactorVAE models in disentanglement_lib achieves 0.779+/-0.094 on the FactorVAE metric, versus 0.764+/-0.075 from our re-run results). We believe these results are consistent with each other. More importantly, we follow the same evaluation parameter setting for the proposed VAE ensemble as other SOTA models, thus are confident with the reported results.
> >
> > Regarding the visualization, it is commonly used as a qualitative measurement where the results are interpreted based on subjective judgement. Using the visualization results to evaluate models can be useful but does not tell the complete story. For example, as we show in Fig. 11, the visualization by TC-VAE shows obvious artifacts despite the high evaluation metrics. Such traversal artifacts can also been seen in other models.
> >
> > We would like to highlight the added ablation results in Fig. 8(b) since last response with the reviewer. To support our discussion on the effect of the cross-model reconstruction, we estimated the DtO of the ablation models and the result is shown in Fig 8(b). For the models without the cross-model reconstruction, we see that the linear transformations among models indeed are close to trivial transformations (signed permutation). More importantly, we show that adding the cross-model reconstruction can further reduce the DtO of the linear transformations among the models in the ensemble. This result can support our intuitive justification that the cross-model objective encourages entangled models to align to disentangled models.

---

### Official Review · AnonReviewer3 · 2020-10-28
**Simple ensembling technique to improve disentanglement**

**Rating:** 5
**Confidence:** 3

**Review:**

This paper proposes a simple and effective technique to improve disentanglement by coupling the latent spaces of different VAE models. It builds on Duan et al. (2019)’s proposed method to rank the representations of different models. By learning a VAE ensemble with linear transformations between the latent spaces and an additional “cross-model” reconstruction loss, the authors show that they can achieve significantly better disentangling.

Strengths:
- The theoretical justification seems reasonable and builds on previous work.
- The experiments are organized to answer three meaningful questions. The results do suggest the VAE ensemble learns better latent representations which can be converted between models with simple, orthogonal linear transformations.

Questions:
- Regarding the last term of the loss in equation (2): for a fixed i and j, the loss is E_{q(z_ij|x)} ||z_jj - z_ij|| = E_{q(z_ij|x)}||z_jj - M_ji z_ii||. This loss term can be optimized by tuning the parameters of VAE i, VAE j, and M_ji. Do you backprop through all these? Or is there a stopgradient on z_ii when used in computing this loss term (i.e. no gradients through VAE i from this loss term)?
- What would be the effect of training the VAE models in two stages: independently first and then jointly in the ensemble? Would it help or hurt disentangling?
- How would you express the total information cost of representing an image across the VAEs in the ensemble (say if you wanted to to compare the information rate to a single VAE)? It doesn't make sense to add up the KL costs linearly.

Suggestions:
- It would help enormously to strengthen the findings and assertions regarding the effect of ensemble size and the gamma hyperparameter.
- Consider adding another disentanglement metric e.g. MIG.
- Figure 5 in the Appendix shows a larger effect on DtO of the number of dims than the gamma hyperparameter. This result (and other results on CelebA) are perhaps worth describing in the main paper.

Minor:
- In Figure 2(a) I assume the curves are overlapping? Does it help to use a log scale for the y-axis?
- How are the latent dimensions sorted in Figure 3?
- Are the scores in Table 2 across different training runs?

---

> ### Author Response · Authors · 2020-11-18
> **Response to AnonReviewer3**
>
> Thank the reviewer for comments! Below we answer each question from the reviewer.
>
> •	The proposed VAE ensemble is trained end-to-end where all parameters of VAE i, VAE j, and M_ji are updated simultaneously. For analysis purpose, the discussion in Section 4.1 assume fixed parameter in order to study the stochastic loss of the VAE ensemble loss.
>
> •	The reviewer raised an interesting question. We believe pertaining individual VAE should help with the VAE ensemble., if not hurting the performance. This is because the individual VAE model do have learn well disentangled representation based on the results in Table 1. Also, the DtO estimation between well-trained individual VAE models in Figure 6 (original Figure 2) shows that the latent representations among these models do not align with each other. There might be a chance for all models in the ensemble to converge to one of the pertained models. That being said, confirming this would require large experiment and we leave it for future work.
>
> •	The VAE ensemble generates multiple VAE models where each individual VAE learns similar disentangled representation. To compare the information rate, one would compare the individual VAE in an ensemble with a single VAE.
>
> •	We agree with the reviewer that the discussion on the effect of ensemble size and the gamma hyperparameter is important. To better understand these effects, we have conducted ablation study as well as conducted further experiments. The results in Table 1 shows that when gamma = 1, the performance of VAE ensemble increases as the ensemble size increases, indicated by the higher mean and smaller variance of the metrics. However, when gamma value is high, the linear transformation loss would force closer mapping between the encoders and reduce the effect of the cross-model reconstruction error of the decoders. As discussed in Section 4.2, the cross-model reconstruction can provide important semantic alignment of the latent representations. We verify its effect by conducting the ablation study as shown in Figure 4.
>
> •	We extend our experiments to include more SOTA models including TC-VAE (Isolating sources of disentanglement in variational autoencoders. Chen et al., 2018) and two versions of DIP-VAE (Variational inference of disentangled latent concepts from unlabeled observations. Kumar et al., 2017).  We also use two widely used supervised metrics including FactorVAE metric and DCI Disentanglement scores (A framework for the quantitative evaluation of disentangled representations. Eastwood & Williams, 2018) as the quantitative measurements.   They are shown to correlate with other common supervised metrics (Challenging Common Assumptions in the Unsupervised Learning of Disentangled Representations. Locatello et al., 2018).  For example, FactorVAE metric and β-VAE metric (Higgins et al., 2017) capture similar notions, while DCI Disentanglement and Mutual Information Gap (MIG) capture similar notions.  In addition, DCI Disentanglement is closely related to the unsupervised model selection method UDR (Unsupervised model selection for variational disentangled representation learning. Duan et al. 2019).
>
> •	We agree with the reviewer that the effect of the latent dimension is important. We extended the discussion on this point. Due to the space limitation, we chose to leave the detail discussion in the appendix but added short sentence in the main paper.
>
> •	We have updated Figure 6.b (Original Figure 2.b) with log scale and achieves better visualization results.
>
> •	For easy visualization, the latent damson in Figure 3 is sorted based on the ‘polarized regime’ estimation metric.
>
> •	Table 2 aims to demonstrate the alignment of the individual models in an ensemble in term of the metric. Given a VAE ensemble, we estimate the metric for each individual model and calculate their standard deviation. Since we test each VAE ensemble multiple runs, an average of these standard deviation is reported.

---

> > ### Author Response · Authors · 2020-11-24
> > **Joint training versus separate training of VAE ensemble**
> >
> > We would like to follow up with the reviewer’s question regarding the separate training versus joint training of the VAE ensemble. We are conducting experiments of separate training of the VAE ensemble for all settings, where each VAE model is first trained and then added to the VAE ensemble. While this effort is still ongoing, based on the current results, we see little difference between separate training and joint training except for ensemble size 2. Specifically, we have the following results for gamma = 1:
> >
> > FactorVAE metric:
> >
> >                                Models  |  Joint training  | Separate training
> >                                VAE_E_2 | 0.711+/-0.106   | 0.788+/-0.039
> >                                VAE_E_3 | 0.794+/-0.030  | 0.795+/-0.043
> >                                VAE_E_4 | 0.833+/-0.037  | 0.825+/-0.040
> >                                VAE_E_5 | 0.828+/-0.016  | 0.772+/-0.036
> >
> > DCI-D metric:
> >
> >                                Models  |  Joint training  | Separate training
> >                                VAE_E_2 | 0.176+/-0.043  | 0.288+/-0.053
> >                                VAE_E_3 | 0.214+/-0.064  | 0.239+/-0.014
> >                                VAE_E_4 | 0.240+/-0.059  | 0.238+/-0.013
> >                                VAE_E_5 | 0.242+/-0.032  | 0.258+/-0.084
> >
> > VAE ensemble with size 2 seems to benefit from separate training, but we did notice that the pertained VAE models in VAE ensemble size 2 can achieve good metrics by themselves. We believe the quality of the pre-trained VAE models play an important role since they do vary a lot in terms of the evaluation metrics. Further investigation and experiments are necessary to support our conjectures mentioned in then earlier response and to conclude the effect of separate training. We are continuing with this effort, and will update the paper when all the results are available as a dedicated section in the Appendix.

---

### Official Review · AnonReviewer4 · 2020-10-30
**Interesting VAE ensemble approach for improving disentangled representations with theoretical as well as experimental validation**

**Rating:** 7
**Confidence:** 4

**Review:**

### Summary:
This submission proposes an ensemble framework to improve learning disentangled representations with Variational Autoencoders (VAEs). The approach builds on the assumption that entangled latent representations learned by VAEs show some “uniqueness” in their latent space structure, while disentangled representations exhibit some “similarity”; an assumption corroborated by recent studies. On that basis, a VAE ensemble approach is proposed where several VAEs are connected through linear mappings between the individual latent spaces to encourage alignment of latent representations and thus disentanglement. A formal derivation of the framework is provided and the formal validity of the underlying assumption demonstrated. Furthermore, empirical evaluation of the proposed approach in comparison to the standard VAE, beta-VAE and FactorVAE on the datasets dSprites (main results, main text) and CelebA (appendix) is performed, yielding improved results on the FactorVAE disentanglement metric (all baseline methods considered) as well as the Distance to Orthogonality (DtO) metric (only standard VAE considered).

### Strengths:
- Significance / Novelty: The proposed approach builds on recent work by Rolinek et al. and Duan et al., which show PCA-like behaviour in VAEs and leverage these results to develop disentanglement scores for model selection. This submission uses these insights for training an ensemble of VAEs in order to improve learning of disentangled representations. The claim is validated both formally as well as empirically on a benchmark dataset (dSprites) and state-of-the-art methods like FactorVAE, where the proposed framework performs favourably. To my knowledge the proposed idea is novel and simple yet potentially quite powerful. This approach could be relevant for other disentanglement methods and a wider audience employing VAE approaches.
- Technical Quality: An important contribution of this paper is the thorough formal derivation and theoretical justification of the approach which to me appears sound. The experimental evaluation is well-designed and mostly succeeds in justifying the claims, with some exceptions outlined below. I believe that all the relevant details to reproduce the results are provided.
- In particular, the results that the DtO comes close to 0 (fig. 2) for the ensemble approach illustrate that the latent representations of the different VAE in the ensemble converge (question 1), i.e. the linear transformations between latent space converge to (signed) permutations. This means that it should not matter which latent representations in the ensemble is studied (in the paper the first model in the ensemble is chosen; lines 274-275). However, I am curious whether the authors considered the results (“polarisation” and FactorVAE scores) for other latent representations (i.e. not the first model) and how much the results agreed?
- Clarity: I consider this paper well-written and well-structured. Relevant details and formal justifications are provided in an appropriate manner resulting in a self-contained paper.

### Weaknesses:
- The ensemble approach comes at a cost which is probably the reason why only up to 5 parallel models were used. Can the authors comment on the running time and memory requirements compared to the competing methods? I think the quality of the paper could be improved if these details and the restrictions of the ensemble approach were provided.
- The results in table 1 (comparison of baseline methods and ensemble approach w.r.t. FactorVAE metric) show that an ensemble of size >=3 can outperform state-of-the-art methods like FactorVAE on the considered FactorVAE metric. However, they also show that it might not always be beneficial to put more weight onto enforcing aligned latent representations for the same ensemble size (gamma > 1). This is a bit at odds with the premise of the paper. As the discussion points out (question 3, lines 285-289), this could be due to balancing different contributions in the more extensive objective function. However, this could also hint at potential optimisation problems for more challenging tasks.
- The examples for the latent traversal (in the appendix) are slightly less convincing and a comparison is only done w.r.t. a standard VAE. However, it would be much more insightful to compare the ensemble approach to beta-VAE and FactorVAE latent traversal results.
- Similar to the last point, in figure 2, it would be quite insightful to see the DtO results for the beta-VAE and especially the FactorVAE. In my opinion, this is a crucial aspect which so far is missing and could justify the approach even more. Isn’t the whole motivation that beta-VAE and FactorVAE should perform slightly worse w.r.t DtO?

### Additional Feedback:
- Figure 1: I like the illustration, however I do not understand the bar plot (“VAE, BetaVAE, FactorVAE, VAE Ensemble”). Maybe an additional annotation could help?
- Line 8: *“sometime”* -> *”sometimes”*
- Line 24: *”state-of-the-arts”* -> *”state-of-the-art”*
- Line 25: *”[…] deploy Variational Autoencoder […]”* -> *”[…] deploy the Variational Autoencoder […]”* or *”[…] deploy Variational Autoencoders […]”*
- Line 37, line 190, line 221 : *”On contrary, […]”* -> *”On the contrary, […]”*
- Line 74: *”[…] closely approximate prior […]”* -> *”[…] closely approximate the prior […]”*
- Line 127: *”[…] models […]”* -> *”[…] model […]”*
- Line 164: *”[…] decomposition L2 term […]”* -> *”[…] decomposition, the L2 term […]”*
- Line 224: *”Such gap […]”* -> *”Such a gap […]”*
- Line 225: *”[…] such case […]”* -> *”[…] such a case […]”*
- Line 233: *”Does VAE ensemble improves […]”* -> *”Does the VAE ensemble improve […]”*

### Recommendation:
This submission was an enjoyable read, it provides some new insights and I believe this paper can pose an important contribution in areas which are concerned with learning disentangled representations and VAE models. In my opinion, the claims of the paper are justified both theoretically and empirically. However, there are certain aspects and concerns outlined above which need to be addressed adequately to warrant a publication. At the moment, I am inclined to accept the paper, but I would like the authors to clarify the concerns and questions above.

### Post-Rebuttal:
I would like to thank the authors for the insightful rebuttal! The authors were able to address my concerns adequately and I believe that the revision improved the quality of the paper quite a bit. Therefore, I stand with my initial recommendation and due to the reasons stated above, I endorse accepting this paper.


### References:
- Rolinek et al., “Variational autoencoders pursue pca directions (by accident)”, CVPR 2019.
- Duan et al., “Unsupervised model selection for variational disentangled representation learning”, ICLR 2019.

---

> ### Author Response · Authors · 2020-11-18
> **Response to AnonReviewer4**
>
> We thank the reviewer for comments! Below we answer each question from the reviewer.
>
> •	The reviewer raised a valid point on the increased computation complexity of the proposed ensemble model. Comparing to training n original VAEs, the proposed VAE ensemble requires additional n*(n-1) linear layers. While this addition does not increase the size of the model much, the linear transformations and the cross-model reconstruction components grow with n*(n-1), which may be computationally expensive especially when n is large. That being said, the results in Section 5 show that the VAE ensemble achieves more stable results comparing to the current state-of-the-art models. Also, its computation is highly parallelizable. This important discussion is added to the paper (line 146-153).
>
> •	The results in Table 1 shows that when gamma = 1, the performance of VAE ensemble increases as the ensemble size increases, indicated by the higher mean and smaller variance of the metrics. However, when gamma value is high, the linear transformation loss would force closer mapping between the encoders and reduce the effect of the cross-model reconstruction error of the decoders. As discussed in Section 4.2, the cross-model reconstruction can provide important semantic alignment of the latent representations. We verify its effect by conducting the ablation study as shown in Figure 4. As a result, balancing the within and between model loss is important for achieving good performance and this can indeed pose a challenge on the optimization. However, this is an issue of all existing state-of-the-art models where it is a common practice to train a number of seeds per hyperparameter setting.
>
> •	The representation by VAE ensemble encourages more “active” latent variables, thus can capture a decomposition of the ground truth generative factors. In Figure 3, we observe that some latent variables in the VAE ensemble are in-between “active” and “passive” modes. This suggests that the VAE ensemble model generates input-dependent factors based on the input complexity. The traversals in Figure 11 (original Figure 8) shows that an ellipse shape does not lead to an “active” latent variable.  However, both heart and square shape lead to an “active” mode of the second latent variable that changes the output. Following the reviewer’s suggestion, we included the latent traversal results from beta-VAE and FactorVAE, as well as other state-of-the-art models that are added to our evaluation. As shown in Figure 11, all these models do not have the decomposition behavior of VAE ensemble where the “active” modes are consistent across different input data.
>
> •	We agree with reviewer that estimating the DtO for beta-VAE and FactorVAE should be insightful. In Figure 2 of the updated paper, we show the comparison of the DtO between the VAE ensemble and other state-of-the-art VAE models. As seen in the figure, the VAE ensemble models with different ensemble size all approach to trivial transformations between the individual models, while other VAE models do not have such property.
>
> •	Following the reviewer’s suggestion, we have updated Figure 1 with annotations on the bar plot to improve its readability.
>
> •	We thank the reviewer for pointing out the grammar errors in the paper! We have corrected them accordingly.

---

> > ### Author Response · Authors · 2020-11-24
> > **The consistency of the models in the same ensemble**
> >
> > One thing we would like to add to the rebuttal for the reviewer is regarding the comment on the consistency of the models in the same ensemble. The reviewer asked whether the results are consistent (“polarisation” and FactorVAE scores) among the latent representations (i.e. not the first model) and how much the results agreed? We can answer this question by referring to results shown in Fig. 3 and Table 2. In Fig. 3, we show the “polarized regime” estimation for individual models in the VAE ensemble with size 2 (left) and 3 (right), where ‘Ensemble i’ stands for the i-th model in the ensemble. We see that individual models do work in similar “polarized regime” and this is also reflected in the consistent FactorVAE metric performance in Table 2. Specifically in Table 2, we estimate the metrics with each individual model in the ensemble and calculate the  standard deviation among them. This calculation is done with 10 different runs and the average of the standard deviation is reported. The small standard deviation suggests that the individual model in the ensemble do achieve similar performance. For FactorVAE metric, we also observe that as ensemble size increases, this consistency decreases. We believe it is due to the difficulty of balancing the within and between model losses. In addition, higher gamma value also lead to reduced consistency. We believe this is due to the reduced effect of cross-model reconstruction that enhances the semantic consistency of the latent representations.

---

### Author Response · Authors · 2020-11-18
**Collective response to reviewers**

We appreciate all three reviewers for taking the time to carefully assessing our work and providing feedbacks!

Following the reviewers’ suggestions, we extensively extended our experiment to demonstrate the effectiveness of the proposed VAE ensemble. These include:
1. we conduct additional experiments by comparing our method with more SOTA models (β-VAE, FactorVAE, TC-VAE and two versions of DIP-VAE) using two metrics (FactorVAE metric and DCI Disentanglement score);
2. we add ablation study to understand the effect of the linear transformation loss and cross-model reconstruction loss;
3. we add the discussion on computation complexity of proposed VAE ensemble.
4. along with these new results, we added discussion on the effect of ensemble size of hyper-parameter gamma.

The paper is updated accordingly to reflect these changes.

---

### Decision · Program_Chairs · 2021-01-07
**Final Decision**

**Decision:**

Reject

**Comment:**

This paper proposes to use an ensemble of VAEs to learn better disentangled representations by aligning their representations through additional losses. This training method is based on recent work by Rolinek et al (2019) and Duan et al (2020), which suggests that VAEs tend to approximate PCA-like behaviour when they are trained to disentangle. The method is well justified from the theoretical perspective, and the quantitative results are good. Saying this, the reviewers raised concerns about the qualitative nature of the learnt representations, which do not look as disentangled as the quantitative measures might suggest. There was a large range of scores given to this paper by the reviewers, which has generated a long discussion. I have also personally looked at the paper. Unfortunately I have to agree that the latent traversal plots do not look as disentangled as the metric scores would suggest, and as one might hope to see on such toy datasets as dSprites. The traversals are certainly subpar to even the most basic approaches to disentanglement, like beta-VAE. For this reason, and given the reviewer scores, I unfortunately have to recommend to reject the paper this time around, however I hope that the authors are able to address the reviewers' concerns and find the source of disagreement between their qualitative and quantitative results for the future revisions of this work.